# SIMPLE REFLOW: IMPROVED TECHNIQUES FOR FAST FLOW MODELS

**Beomsu Kim**
Apple and KAIST

**Yu-Guan Hsieh**
Apple

**Michal Klein**
Apple

**Marco Cuturi**
Apple

**Jong Chul Ye**
KAIST

**Bahjat Kawar**
Apple

**James Thornton**
Apple

## ABSTRACT

Diffusion and flow-matching models achieve remarkable generative performance but at the cost of many sampling steps, this slows inference and limits applicability to time-critical tasks. The ReFlow procedure can accelerate sampling by straightening generation trajectories. However, ReFlow is an iterative procedure, typically requiring training on simulated data, and results in reduced sample quality. To mitigate sample deterioration, we examine the design space of ReFlow and highlight potential pitfalls in prior heuristic practices. We then propose seven improvements for training dynamics, learning and inference, which are verified with thorough ablation studies on CIFAR10 $32 \times 32$, AFHQv2 $64 \times 64$, and FFHQ $64 \times 64$. Combining all our techniques, we achieve state-of-the-art FID scores (without / with guidance, resp.) for fast generation via neural ODEs: 2.23 / 1.98 on CIFAR10, 2.30 / 1.91 on AFHQv2, 2.84 / 2.67 on FFHQ, and 3.49 / 1.74 on ImageNet-64, all with merely 9 neural function evaluations.

## 1 INTRODUCTION

The diffusion model (DMs) paradigm (Sohl-Dickstein et al., 2015; Ho et al., 2020) has changed the landscape of generative modelling of perceptual data, benefitting from scalability, stability and remarkable performance in a diverse set of tasks ranging from unconditional generation (Dhariwal & Nichol, 2021) to conditional generation such as image restoration (Chung et al., 2023), editing (Meng et al., 2022), translation (Su et al., 2023), and text-to-image generation (Rombach et al., 2022). However, to generate samples, DMs require numerically integrating a differential equation using tens to hundreds of neural function evaluations (NFEs) (Song et al., 2021b;a). Naively reducing the NFE increases discretization error, causing sample quality to worsen. This has sparked wide interest in accelerating diffusion sampling (Song et al., 2021a; Lu et al., 2022; Zhang & Chen, 2023; Kim & Ye, 2023; Salimans & Ho, 2022; Song et al., 2023).

Flow matching models (FMs) (Lipman et al., 2023) are a closely related class of generative model sharing similar training and sampling procedures and enjoying similar performance to diffusion models. Indeed, FM and DMs coincide for a particular choice of forward process (Kingma & Gao, 2023), and are also related to stochastic interpolants (Albergo & Vanden-Eijnden, 2023; Albergo et al., 2023). Whereas diffusion models relate to entropically regularized transport (Bortoli et al., 2021; Shi et al., 2023; Peluchetti, 2023), a key property of flow matching models is their connection to non-regularized optimal transport, and hence deterministic, straight trajectories (Liu, 2022).

While there exist a plethora of acceleration techniques, one promising, yet less explored avenue is ReFlow (Liu et al., 2022; Liu, 2022), also known as Iterative Markovian Fitting (IMF) (Shi et al., 2023). ReFlow straightens ODE trajectories through flow-matching between marginal distributions coupled by a previously trained flow ODE, rather than using an independent coupling. Theoretically, with an infinite number of ReFlow updates, the resulting learned ODE should be straight, which enables perfect translation between the marginals with a single function evaluation (Liu et al., 2022).

In practice however, ReFlow results in a drop in sample quality (Liu et al., 2022; 2024). To address this problem, recent works on sampling acceleration via ReFlow opt to use heuristic tricks such as perceptual losses that only loosely adhere to the underlying theory (Lee et al., 2024; Zhu et al.,

2024). Consequently, it is unclear whether the marginals are still preserved after ReFlow. This is problematic, as exact inversion and tractable likelihood calculation require access to a valid probability flow ODE between the marginals. Moreover, these two functions are critical to downstream applications such as zero-shot classification (Li et al., 2023) etc.

The goal of this work is to study and mitigate the performance drop after ReFlow without violating the theoretical setup. Although technical in nature, we call our method *simple*, as, similar to *simple diffusion* (Hoogeboom et al., 2023), it does not rely on latent-encoders, perceptual losses, or premetrics, whose effect on the learned marginals is poorly understood. To this end, we first disentangle the components of ReFlow. Next, we examine the pitfalls of previous practices. Finally, we propose enhancements within the theoretical bounds, and verify them through rigorous ablation studies.

Our contributions are summarized as follows.

- **We generalize and categorize the design choices of ReFlow (Section 3.1).** We generalize the ReFlow training loss and categorize the design choices of ReFlow into three key groups: *training dynamics*, *learning*, and *inference*. Within each group, we discuss previous practices, highlight their potential pitfalls, and propose improved techniques.
- **We analyze each improvement via ablations (Sections 3.2, 3.3, and 3.4).** For each proposed improvement, we verify its effect on sample quality via extensive ablations on three datasets: CIFAR10 $32 \times 32$ (Krizhevsky, 2009), FFHQ $64 \times 64$ (Karras et al., 2019), and AFHQv2 $64 \times 64$ (Choi et al., 2020). We demonstrate that our techniques are robust, and they offer consistent gains in FID scores (Heusel et al., 2017) on all three datasets.
- **We achieve state-of-the-art results (Section 4).** With all our improvements, we set state-of-the-art FIDs for fast generation via neural ODEs, without perceptual losses or premetrics. Our best models achieve 2.23 FID on CIFAR10, 2.30 FID on AFHQv2, 2.84 FID on FFHQ, and 3.49 FID on ImageNet-64, all with merely 9 NFEs. In particular, our models outperform the latest fast neural ODEs such as curvature minimization (Lee et al., 2023) and minibatch OT flow matching (Pooladian et al., 2023). We are also able to further enhance the perceptual quality of samples via guidance, setting 1.98 FID on CIFAR10, 1.91 FID on AFHQv2, 2.67 FID on FFHQ, and 1.74 FID on ImageNet-64, also with 9 NFEs.

## 2 BACKGROUND

Let $\mathbb{P}_0$ and $\mathbb{P}_1$ be two data distributions on $\mathbb{R}^d$. Rectified Flow (RF) (Liu et al., 2022; Liu, 2022) is an algorithm which learns straight ordinary differential equations (ODEs) between $\mathbb{P}_0$ and $\mathbb{P}_1$ by iterating a procedure called ReFlow. Below, we describe ReFlow, and explain how it can be applied to diffusion probability flow ODEs to learn fast generative flow models.

### 2.1 FLOW MATCHING AND REFLOW

Let us first define the flow matching (FM) loss and its equivalent formulation as a denoising problem

$$\mathcal{L}_{\mathrm{FM}}(\theta; \mathbb{Q}_{01}) := \mathbb{E}_{(\boldsymbol{x}_0, \boldsymbol{x}_1) \sim \mathbb{Q}_{01}} \mathbb{E}_{t \sim \mathrm{unif}(0,1)} \left[ \ell_{\mathrm{MSE}}(\boldsymbol{x}_1 - \boldsymbol{x}_0, \boldsymbol{v}_\theta(\boldsymbol{x}_t, t)) \right] \tag{1}$$

$$:= \mathbb{E}_{(\boldsymbol{x}_0, \boldsymbol{x}_1) \sim \mathbb{Q}_{01}} \mathbb{E}_{t \sim \mathrm{unif}(0,1)} [t^{-2} \cdot \ell_{\mathrm{MSE}}(\boldsymbol{x}_0, \boldsymbol{D}_\theta(\boldsymbol{x}_t, t))] \tag{2}$$

where $\boldsymbol{v}_\theta : \mathbb{R}^d \times (0, 1) \to \mathbb{R}^d$ is a velocity parameterized by $\theta$, $\ell_{\mathrm{MSE}}(\boldsymbol{x}, \boldsymbol{y}) := \|\boldsymbol{x} - \boldsymbol{y}\|_2^2$, $\boldsymbol{x}_t := (1 - t)\boldsymbol{x}_0 + t\boldsymbol{x}_1$, and $\mathbb{Q}_{01}$ is a coupling, *i.e.*, a joint distribution, of $\mathbb{P}_0$ and $\mathbb{P}_1$, and $\boldsymbol{D}_\theta(\boldsymbol{x}_t, t) := \boldsymbol{x}_t - t\boldsymbol{v}_\theta$ is a denoiser that is optimized to recover the original data $\boldsymbol{x}_0$ given a corrupted observation $\boldsymbol{x}_t$ and time $t$ as inputs. According to the FM theory, a velocity which minimizes Eq. (1) or a denoiser which minimizes Eq. (2) can translate samples from $\mathbb{P}_i$ to $\mathbb{P}_{1-i}$, $i \in \{0, 1\}$, by solving the ODE

$$d\boldsymbol{x}_t = \boldsymbol{v}_\theta(\boldsymbol{x}_t, t) \, dt = t^{-1}(\boldsymbol{x}_t - \boldsymbol{D}_\theta(\boldsymbol{x}_t, t)) \, dt, \quad t \in (0, 1) \tag{3}$$

from $t = i$ to $1 - i$ (Lipman et al., 2023). Let us call the denoiser which minimizes Eq. (2) w.r.t. the independent coupling $\mathbb{Q}_{01} = \mathbb{P}_0 \otimes \mathbb{P}_1$ as $\boldsymbol{D}_\theta^1$.

For $n \geq 1$, ReFlow minimizes Eq. (2) with coupling induced by $\boldsymbol{D}_\theta^n$ to obtain $\boldsymbol{D}_\theta^{n+1}$ whose ODE has a lower transport cost. Specifically, observe that Eq. (3) with $\boldsymbol{D}_\theta^n$ induces a coupling

$$d\mathbb{Q}_{01}^n(\boldsymbol{x}_0, \boldsymbol{x}_1) := \begin{cases} d\mathbb{P}_1(\boldsymbol{x}_1)\delta(\boldsymbol{x}_0 - \mathrm{solve}(\boldsymbol{x}_1, \boldsymbol{D}_\theta^n, 1, 0)) \\ d\mathbb{P}_0(\boldsymbol{x}_0)\delta(\boldsymbol{x}_1 - \mathrm{solve}(\boldsymbol{x}_0, \boldsymbol{D}_\theta^n, 0, 1)) \end{cases} \tag{4}$$

where $\delta$ is the Dirac delta, and $\text{solve}(\boldsymbol{x}, \boldsymbol{D}_\theta^n, t_0, t_1)$ solves Eq. (3) from time $t = t_0$ to $t_1$ with initial point $\boldsymbol{x}$. Concretely, given $\boldsymbol{x}_i \sim \mathbb{P}_i$ for $i \in \{0, 1\}$, we sample $\boldsymbol{x}_{1-i} \sim \mathbb{Q}_{1-i|i}(\cdot|\boldsymbol{x}_i)$ by integrating Eq. (3) from $t = i$ to $1 - i$ starting from $\boldsymbol{x}_i$. The two expressions in Eq. (4) are equivalent, since an ODE defines a bijective map between initial and terminal points.

RF guarantees that if $\boldsymbol{D}_\theta^{n+1}$ is a minimizer of $\mathcal{L}_{\text{FM}}(\theta; \mathbb{Q}_{01}^n)$, the ODE with $\boldsymbol{D}_\theta^{n+1}$ converges to a perfectly straight ODE as $n \to \infty$. If an ODE has perfectly straight trajectories, it is possible to translate between the marginals with a single Euler step, *e.g.*, $\boldsymbol{x}_0 = \boldsymbol{D}_\theta(\boldsymbol{x}_1, 1)$ when translating from $t = 1$ to $0$ (see Section 3 of Liu et al., 2022).

## 2.2 ReFlow with Diffusion Probability Flow ODEs

Given distribution $\mathbb{P}_0$ on $\mathbb{R}^d$ and Gaussian perturbation kernel $d\mathbb{Q}_{\sigma|0}(\boldsymbol{y}_\sigma|\boldsymbol{y}_0) \coloneqq \mathcal{N}(\boldsymbol{y}_\sigma|\boldsymbol{y}_0, \sigma^2\boldsymbol{I})$, DMs solve the denoising score matching (DSM) (Vincent, 2011) problems

$$\mathcal{L}_{\text{DSM}}(\theta) \coloneqq \mathbb{E}_{\sigma \sim \mathbb{S}} \mathbb{E}_{\boldsymbol{y}_0 \sim \mathbb{P}_0} \mathbb{E}_{\boldsymbol{y}_\sigma \sim \mathbb{Q}_{\sigma|0}(\cdot|\boldsymbol{y}_0)} \left[ \ell_{\text{MSE}}(\boldsymbol{y}_0, \boldsymbol{F}_\theta(\boldsymbol{y}_\sigma, \sigma)) \right] \tag{5}$$

to learn a denoiser $\boldsymbol{F}_\theta : \mathbb{R}^d \times (0, \infty) \to \mathbb{R}^d$. $\boldsymbol{F}_\theta$ then defines a probability flow ODE between $\mathbb{P}_0$ and $\mathbb{P}_0 * \mathcal{N}(\boldsymbol{0}, \hat{\sigma}^2\boldsymbol{I}) \approx \mathcal{N}(\boldsymbol{0}, \hat{\sigma}^2\boldsymbol{I})$ for a large $\hat{\sigma}$:

$$d\boldsymbol{y}_\sigma = \sigma^{-1}(\boldsymbol{y}_\sigma - \boldsymbol{F}_\theta(\boldsymbol{y}_\sigma, \sigma)) \, d\sigma, \quad \sigma \in (0, \infty). \tag{6}$$

When $\mathbb{P}_1$ is standard normal, Eq. (3) with $\boldsymbol{D}_\theta^1$ and Eq. (6) are equivalent, as Eq. (3) with the change of variables $(\boldsymbol{y}_\sigma, \sigma) \coloneqq (\frac{\boldsymbol{x}_t}{1-t}, \frac{t}{1-t})$ and Eq. (6) are identical (Lee et al., 2024). It follows that we can straighten diffusion probability flow ODE trajectories via ReFlow. Specifically, with the coupling

$$d\mathbb{Q}_{01}^1(\boldsymbol{x}_0, \boldsymbol{x}_1) = \begin{cases} d\mathbb{P}_1(\boldsymbol{x}_1)\delta(\boldsymbol{x}_0 - \text{solve}(\frac{\boldsymbol{x}_1}{1-t}, \boldsymbol{F}_\theta, \frac{t}{1-t}, 0)) \\ d\mathbb{P}_0(\boldsymbol{x}_0)\delta(\boldsymbol{x}_1 - (1-t) \cdot \text{solve}(\boldsymbol{x}_0, \boldsymbol{F}_\theta, 0, \frac{t}{1-t})) \end{cases} \tag{7}$$

where $t \approx 1$ and $\text{solve}(\boldsymbol{y}, \boldsymbol{F}_\theta, \sigma_0, \sigma_1)$ solves Eq. (6) from $\sigma = \sigma_0$ to $\sigma_1$ with initial point $\boldsymbol{y}$, we can minimize Eq. (1) to learn $\boldsymbol{D}_\theta^2$, and so on. Because optimizing Eq. (1) is often expensive, a typical procedure is to perform one ReFlow step with Eq. (7) to get $\boldsymbol{D}_\theta^2$, and distill Eq. (3) trajectories into a student model for one-step generation (Liu et al., 2022; Zhu et al., 2024; Liu et al., 2024).

## 3 Improved Techniques for ReFlow

We now investigate the design space of ReFlow and propose improvements. Specifically, in Section 3.1, we generalize the FM loss and identify the components that constitute ReFlow. The components are organized into three groups – *training dynamics*, *learning*, and *inference*. In Sections 3.2, 3.3, and 3.4, we investigate the pitfalls of previous practices and propose improved techniques in each group. To show that our improvements are robust, we provide rigorous ablation studies on CIFAR10 $32 \times 32$, AFHQv2 $64 \times 64$, and FFHQ $64 \times 64$. *We find that ReFlow training and sampling are very different from those of DMs, and generally require distinct techniques for optimal performance.*

## 3.1 The Design Space of ReFlow

**Generalizing weight and time distribution.** Let the joint distribution of $(\boldsymbol{x}_0, \boldsymbol{x}_1, t, \boldsymbol{x}_t)$ be given by $\boldsymbol{x}_0, \boldsymbol{x}_1 \sim d\mathbb{Q}_{01}, t \sim \mathbb{T}$, and $\boldsymbol{x}_t = (1-t)\boldsymbol{x}_0 + t\boldsymbol{x}_1$. Then Eq. (2) can also be expressed as

$$\mathcal{L}_{\text{FM}}(\theta; \mathbb{Q}_{01}) = \mathbb{E}_{t \sim \text{unif}(0,1)} \mathbb{E}_{\boldsymbol{x}_t \sim \mathbb{Q}_t} \left[ w(t) \cdot \mathcal{L}_{\text{FM}}(\theta; \mathbb{Q}_{01}, \boldsymbol{x}_t, t) \right], \tag{8}$$

$$\textit{where} \ \ \mathcal{L}_{\text{FM}}(\theta; \mathbb{Q}_{01}, \boldsymbol{x}_t, t) \coloneqq \mathbb{E}_{\boldsymbol{x}_0 \sim \mathbb{Q}_{0|t}(\cdot|\boldsymbol{x}_t)} \left[ \ell_{\text{MSE}}(\boldsymbol{x}_0, \boldsymbol{D}_\theta(\boldsymbol{x}_t, t)) \right], \tag{9}$$

and $w(t) = t^{-2}$. This shows that the FM loss is separable w.r.t. $(\boldsymbol{x}_t, t)$, and the optimal denoising function is given by the posterior mean (Robbins, 1956): $\boldsymbol{D}^*(\boldsymbol{x}_t, t) = \mathbb{E}_{\boldsymbol{x}_0 \sim \mathbb{Q}_{0|t}(\cdot|\boldsymbol{x}_t)}[\boldsymbol{x}_0]$. Hence, we may replace $w(t)$ with a general weight $w(\boldsymbol{x}_t, t)$ and use a general time distribution $\mathbb{T}$

$$\mathcal{L}_{\text{FM}}(\theta; \mathbb{Q}_{01}) = \mathbb{E}_{t \sim \mathbb{T}} \mathbb{E}_{\boldsymbol{x}_t \sim \mathbb{Q}_t} \left[ w(\boldsymbol{x}_t, t) \cdot \mathcal{L}_{\text{FM}}(\theta; \mathbb{Q}_{01}, \boldsymbol{x}_t, t) \right]. \tag{10}$$

This is minimized under the same condition, given that $w(\boldsymbol{x}_t, t) > 0$ and $\mathbb{T}$ is supported on $(0, 1)$.

|  | RF | RF++ | Baseline | Simple ReFlow (Ours) |
|---|---|---|---|---|
| **Train Dynamics (Sec. 3.2)** |  |  |  |  |
| Weight $w(\boldsymbol{x}_t, t)$ | $1/t^2$ | $1, 1/t$ | $1$ | $1/\operatorname{sg}[\mathcal{L}_{\mathrm{FM}}(\theta; \mathbb{Q}_{01}, \boldsymbol{x}_t, t)]$ |
| Time distribution $d\mathbb{T}(t) \propto$ | $1$ | $\cosh(4(t-0.5))$ | $\cosh(4(t-0.5))$ | $10^t$ |
| Loss function $\ell$ | $\ell_{\mathrm{MSE}}$, LPIPS | Pseudo-Huber, LPIPS | $\ell_{\mathrm{MSE}}$ | $\ell_{\boldsymbol{I}+\lambda\,\mathrm{HPF}}$ |
| **Learning (Sec. 3.3)** |  |  |  |  |
| $\boldsymbol{D}_\theta$ initialization with DM | ✗ | ✓ | ✓ | ✓ |
| $\boldsymbol{D}_\theta$ dropout probability | 0.15 | Equal to EDM | 0.15 | $\ll 0.15$ |
| Sampling from $\mathbb{Q}_{01}$ | Backward | Backward | Backward | Forward, Projection |
| **Inference (Sec. 3.4)** |  |  |  |  |
| ODE Solver | Euler | Euler, Heun | Heun | DPM-Solver |
| Discretization of $[0, 1]$ | Uniform | Uniform | Uniform | Sigmoid $\kappa = 20$ |
| **Reference** | (Liu et al., 2022) (Zhu et al., 2024) | (Lee et al., 2024) | – | – |

Table 1: Comparison of practices for optimizing the ReFlow loss Eq. (13) and solving the ODE Eq. (3). sg means stop gradient and HPF denotes high-pass filter. **Baseline** is the combination of most recent techniques which do not violate the flow matching theory.

**Generalizing the loss function.** We also consider using general loss functions in ReFlow, *i.e.*,

$$\mathcal{L}_{\mathrm{FM}}(\theta; \mathbb{Q}_{01}, \boldsymbol{x}_t, t) = \mathbb{E}_{\boldsymbol{x}_0 \sim \mathbb{Q}_{0|t}(\cdot|\boldsymbol{x}_t)} \left[\ell(\boldsymbol{x}_0, \boldsymbol{D}_\theta(\boldsymbol{x}_t, t))\right] \tag{11}$$

for general $\ell : \mathbb{R}^d \times \mathbb{R}^d \to \mathbb{R}$. It is difficult to precisely characterize the class of $\ell$ that preserves the minimizers of Eq. (1), and popular losses such as LPIPS (Kendall et al., 2018) and pseudo-Huber (PH) (Song & Dhariwal, 2024) lack this guarantee. However, $\ell_{\mathrm{MSE}}$ has been observed to be sub-optimal compared to, *e.g.*, LPIPS and PH for training fast models (Lee et al., 2024). To mitigate this trade-off between theoretical correctness and practicality, we consider a wider class of losses

$$\ell_\phi(\boldsymbol{x}, \boldsymbol{y}) := \|\phi(\boldsymbol{x}) - \phi(\boldsymbol{y})\|_2^2 \tag{12}$$

for invertible linear maps $\phi : \mathbb{R}^d \to \mathbb{R}^d$. This again ensures that the loss is minimized when $\boldsymbol{D}_\theta$ outputs the posterior mean, and $\ell_{\mathrm{MSE}}$ is a special case of this loss with the identity map $\phi = \boldsymbol{I}$.

**Generalized FM loss.** Combining the two generalizations, we have our generalized FM loss

$$\mathcal{L}_{\mathrm{GFM}}(\theta; \mathbb{Q}_{01}) = \mathbb{E}_{t \sim \mathbb{T}} \mathbb{E}_{\boldsymbol{x}_t \sim \mathbb{Q}_t} \left[w(\boldsymbol{x}_t, t) \cdot \mathcal{L}_{\mathrm{GFM}}(\theta; \mathbb{Q}_{01}, \boldsymbol{x}_t, t)\right], \tag{13}$$

$$where \quad \mathcal{L}_{\mathrm{GFM}}(\theta; \mathbb{Q}_{01}, \boldsymbol{x}_t, t) := \mathbb{E}_{\boldsymbol{x}_0 \sim \mathbb{Q}_{0|t}(\cdot|\boldsymbol{x}_t)} \left[\ell_\phi(\boldsymbol{x}_0, \boldsymbol{D}_\theta(\boldsymbol{x}_t, t))\right]. \tag{14}$$

The following proposition ensures its theoretical correctness. Proof is deferred to Appendix E.1.

**Proposition 1.** *Let $w(\boldsymbol{x}_t, t)$, $d\mathbb{T}(t)$ be positive, and $\phi$ be an invertible linear map. Then, $\theta$ minimizes Eq. (13) if and only if it minimizes Eq. (1).*

### 3.1.1 TRAINING DYNAMICS, LEARNING, AND INFERENCE

We now observe that there are seven components that constitute ReFlow: time distribution $\mathbb{T}$, training dataset (empirical realization of $\mathbb{Q}_{01}$), weight $w(\boldsymbol{x}_t, t)$, loss function $\ell_\phi$, denoiser $\boldsymbol{D}_\theta$, and ODE solver and discretization schedule for solving Eq. (3). We categorize them into three groups below.

*Training dynamics* influence the path that the model takes towards the minimizers of Eq. (13) during training. Although the solution to which the model converges may change if dynamics changes, training dynamics do not impact the solution set itself. Weight function $w(\boldsymbol{x}_t, t)$, time distribution $\mathbb{T}$, and loss function $\ell_\phi$ belong here. *Learning* influence the solution set of Eq. (13) by constraining the hypothesis class or by changing the training dataset. Parameterization of $\boldsymbol{D}_\theta$ and how we sample from $\mathbb{Q}_{01}$ belong here. Finally, *inference* influence generation or inversion of samples given a trained model. ODE solver and time discretization of the unit interval belong here.

In Tab. 1, we describe recent ReFlow practices within our framework. Baseline is the collection of most recent ReFlow techniques which do not violate FM theory (Lipman et al., 2023). We will build up improvements on this baseline setting in the subsequent sections.

### 3.2 IMPROVING TRAINING DYNAMICS

**Evaluation protocol.** To evaluate a training setting, we perform a single ReFlow step. Unless written otherwise, we initialize ReFlow denoisers with pre-trained EDM (Karras et al., 2022) denoisers

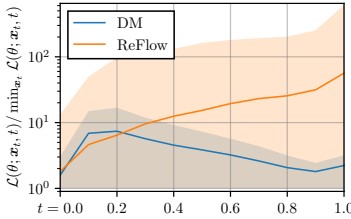

Figure 1: Min., avg., max. relative losses after training on CIFAR10.

| $w(\boldsymbol{x}_t, t)$ | CIFAR10 | AFHQv2 | FFHQ |
|---|---|---|---|
| $1$ | 2.83 | 2.87 | 4.28 |
| $1/t$ | 2.77 | 2.76 | 4.01 |
| $1/t^2$ | 2.76 | **2.74** | 4.04 |
| $(\sigma^2 + 0.5^2)/(0.5\sigma)^2$ | 2.78 | 2.82 | 4.04 |
| $1/\mathbb{E}_{\boldsymbol{x}_t}[\mathrm{sg}[\mathcal{L}_{\mathrm{GFM}}(\theta; \mathbb{Q}_{01}, \boldsymbol{x}_t, t)]]$ | 2.74 | 2.79 | **3.83** |
| $1/\mathrm{sg}[\mathcal{L}_{\mathrm{GFM}}(\theta; \mathbb{Q}_{01}, \boldsymbol{x}_t, t)]$ | **2.61** | **2.74** | **3.83** |

Table 2: Comparison of various $w(\boldsymbol{x}_t, t)$, combined with baseline $\mathbb{T}$, $\ell$, and learning choices. Best numbers are **bolded**, and second best are underlined.

and optimize Eq. (13) with $\mathbb{Q}_{01} = \mathbb{Q}_{01}^1$ of Eq. (7) for $200k$ iterations. We sample $1M$ pairs from $\mathbb{Q}_{01}^1$ by solving Eq. (6) from $t = 1$ to $0$ with EDM models and use them for training. We measure the performance of an optimized model by computing the FID (Heusel et al., 2017) between $50k$ generated images and all available dataset images. Samples are generated by solving Eq. (3) with the Heun solver (Ascher & Petzold, 1998) with 9 NFEs, and we use the sigmoid discretization instead of the baseline uniform discretization for reasons discussed in Appendix F.1. We report the minimum FID out of three random generation trials. See Appendix D for a complete description.

### 3.2.1 LOSS NORMALIZATION

**Previous practice.** There is little study on suitable loss weights for ReFlow training. For instance, Lee et al. (2024) use $w(\boldsymbol{x}_t, t) = 1$, and such choices can be detrimental to training, as the weighted loss $w(\boldsymbol{x}_t, t) \cdot \mathcal{L}_{\mathrm{GFM}}(\theta; \mathbb{Q}_{01}, \boldsymbol{x}_t, t)$ without proper modulation by $w(\boldsymbol{x}_t, t)$ can have vastly different scales w.r.t. $t$, leading to slow and unstable model convergence. Typically, the loss vanishes as $t \to 0$ since $\boldsymbol{x}_t \to \boldsymbol{x}_0$, and to counteract this, previous works have suggested

$$w(\boldsymbol{x}_t, t) = \begin{cases} 1/t & \text{for ReFlow (Lee et al., 2024),} \\ (\sigma^2 + 0.5^2)/(0.5\sigma)^2, \ \sigma := \frac{t}{1-t} & \text{for DMs (Karras et al., 2022),} \\ 1/\mathbb{E}_{\boldsymbol{x}_t \sim \mathbb{Q}_t}[\mathrm{sg}[\mathcal{L}_{\mathrm{GFM}}(\theta; \mathbb{Q}_{01}, \boldsymbol{x}_t, t)]] & \text{for DMs (Karras et al., 2023b),} \end{cases}$$

where $\mathrm{sg}[\cdot]$ is stop-gradient. The FM weight $1/t^{-2}$ in Eq. (2) naturally emphasizes $t \approx 0$ as well.

However, we claim that such weights constant w.r.t. $\boldsymbol{x}_t$ can be sub-optimal for ReFlow, as ReFlow loss scales can vary greatly w.r.t. $\boldsymbol{x}_t$ even for fixed $t$. For instance, the following proposition shows that, at initialization, relative loss for DM at $t = 1$ is constant whereas relative loss for ReFlow can be arbitrarily large. Proof is deferred to Appendix E.2.

**Proposition 2.** *Assume output layer zero initialization for the DM denoiser and DM initialization for the ReFlow denoiser. Then maximum relative losses for DM and ReFlow at $t = 1$ are*

$$\max_{\boldsymbol{x}_1} \mathcal{L}_{\mathrm{DM}}(\theta; \mathbb{P}_0 \otimes \mathbb{P}_1, \boldsymbol{x}_1, 1) / \min_{\boldsymbol{x}_1} \mathcal{L}_{\mathrm{DM}}(\theta; \mathbb{P}_0 \otimes \mathbb{P}_1, \boldsymbol{x}_1, 1) = 1$$

$$\max_{\boldsymbol{x}_1} \mathcal{L}_{\mathrm{GFM}}(\theta; \mathbb{Q}_{01}^1, \boldsymbol{x}_1, 1) / \min_{\boldsymbol{x}_1} \mathcal{L}_{\mathrm{GFM}}(\theta; \mathbb{Q}_{01}^1, \boldsymbol{x}_1, 1) = \max_{\boldsymbol{x}_0, \boldsymbol{x}_0'} \|\boldsymbol{x}_0 - \boldsymbol{\mu}_0\|_2^2 / \|\boldsymbol{x}_0' - \boldsymbol{\mu}_0\|_2^2$$

*resp., where $\boldsymbol{\mu}_0 = \mathbb{E}_{\boldsymbol{x}_0 \sim \mathbb{P}_0}[\boldsymbol{x}_0]$, and $\min_{\boldsymbol{x}_i}, \max_{\boldsymbol{x}_i}$ is taken w.r.t. $\boldsymbol{x}_i$ in the support of $\mathbb{P}_i$.*

In fact, in Fig. 1, we observe that ReFlow loss varies greatly w.r.t. $\boldsymbol{x}_t$ after training as well. In contrast to DM training loss whose minimum and maximum values differ by a factor of at most 20, minimum ReFlow loss is at least $\times 100$ smaller than the maximum loss for all $t > 0.2$.

**Our improvement.** Multi-task learning interpretation of loss normalization (Zhang et al., 2018; Karras et al., 2023b) (see Appendix G) motivates a simple improvement by using

$$w(\boldsymbol{x}_t, t) = 1/\mathrm{sg}[\mathcal{L}_{\mathrm{GFM}}(\theta; \mathbb{Q}_{01}, \boldsymbol{x}_t, t)]. \tag{15}$$

Similar to Karras et al. (2023b), we keep track of the loss values during training with a small neural net that is optimized alongside $\boldsymbol{D}_\theta$ using the parameterization $w(\boldsymbol{x}_t, t) = \exp(-f_\phi(\boldsymbol{x}_t, t))$.

**Ablations.** In Tab. 2, we compare our weight with all aforementioned weights. As expected, the uniform weight $w(\boldsymbol{x}_t, t) = 1$ has the worst performance, as it is unable to account for vanishing loss as $t \to 0$. We get noticeable FID gain by using weights such as $1/t$ or which place a larger emphasis on $t = 0$. Our weight, which accounts for loss variance w.r.t. both $t$ and $\boldsymbol{x}_t$, yields the best FID across all three datasets. The gap between baselines and our weight is especially large on CIFAR10.

### 3.2.2 TIME DISTRIBUTION

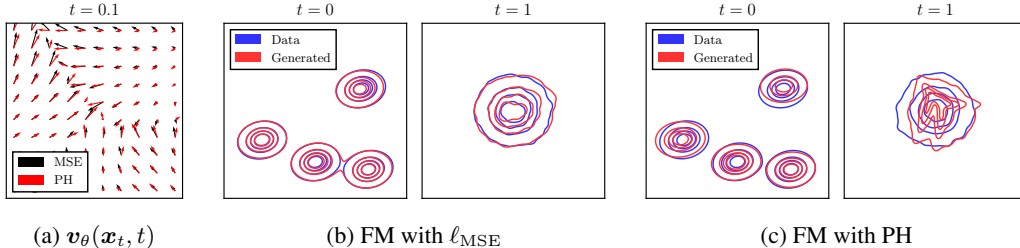

(a) $\boldsymbol{v}_\theta(\boldsymbol{x}_t, t)$       (b) FM with $\ell_{\mathrm{MSE}}$       (c) FM with PH

Figure 3: Comparison of flow matching (FM) with $\ell_{\mathrm{MSE}}$ and Pseudo Huber (PH) losses.

**Previous practice.** Liu et al. (2022) and Zhu et al. (2024) use a uniform distribution on $(0, 1)$. On the other hand, Lee et al. (2024) notice better performance with a time distribution whose density is proportional to a shifted hyperbolic cosine function, *i.e.*, $d\mathbb{T}(t) \propto \cosh(4(t - 0.5))$, which has peaks at $t = 0$ and $1$. The rationale behind using such a distribution is that, as Eq. (3) converges to a straight ODE with ReFlows, the denoiser needs to directly predict data from noise at $t \approx 1$ and vice versa at $t \approx 0$, so it is beneficial to emphasize those regions via $\mathbb{T}$.

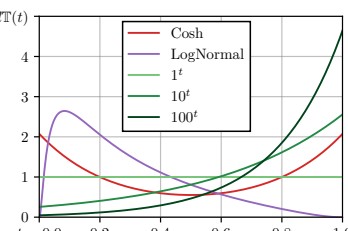

Figure 2: Time distribution densities.

**Our improvement.** The peak at $t = 0$ of the baseline cosh time density compensates for vanishing loss as $t \to 0$, but as we normalize the loss with our weight, this peak is now no longer necessary. Thus, we use a distribution with density proportional to the increasing exponential, *i.e.*, $d\mathbb{T}(t) \propto a^t$ for $a \geq 1$.

**Ablations.** We compare the performance of the exponential distribution with $a \in \{1, 10, 100\}$, where $a = 1$ corresponds to the uniform distribution. We also compare

| $d\mathbb{T}(t) \propto$ | CIFAR10 | AFHQv2 | FFHQ |
|---|---|---|---|
| $\cosh(4(t - 0.5))$ | **2.61** | 2.74 | 3.83 |
| lognormal | 3.28 | 3.20 | 4.48 |
| uniform $(1^t)$ | 2.65 | 2.70 | 3.85 |
| $10^t$ | 2.62 | **2.69** | **3.77** |
| $100^t$ | 2.68 | **2.69** | 3.93 |

Table 3: Comparison of various $\mathbb{T}$, combined with our $w(\boldsymbol{x}_t, t)$ and baseline $\ell$ and learning choices.

with the lognormal distribution, which has been observed to be effective for training DMs and CMs (Karras et al., 2022; 2023b; Song & Dhariwal, 2024) Fig. 2 displays time densities, and Tab. 3 shows training results. We first note that an emphasis on $t = 1$ is necessary, as evidenced by severe FID degradation with the lognormal distribution. We then observe that $10^t$, which closely resembles the cosh distribution, but without the peak at $t = 0$, has consistently good performance, while suffering from a slight loss on CIFAR10. Other choices such as $1^t$ or $100^t$ are either too flat or sharp, yielding worse FID. Hence, we propose to take $d\mathbb{T}(t) \propto 10^t$.

### 3.2.3 LOSS FUNCTION

**Previous practice.** To accelerate convergence and mitigate sample quality degradation during Re-Flow, previous works have employed heuristic losses in Eq. (2) such as LPIPS and

$$\ell(\boldsymbol{x}, \boldsymbol{y}) = \begin{cases} (1/t) \cdot (\|\boldsymbol{x} - \boldsymbol{y}\|_2^2 + (ct)^2)^{1/2} - c & \text{Pseudo-Huber (PH) (Lee et al., 2024),} \\ \mathrm{LPIPS}(\boldsymbol{x}, \boldsymbol{y}) + (1 - t) \cdot \mathrm{PH}(\boldsymbol{x}, \boldsymbol{y}) & \text{LPIPS+PH (Lee et al., 2024).} \end{cases}$$

While such losses perform better in practice than $\ell_{\mathrm{MSE}}$ in terms of FID, they do not ensure Eq. (1) is minimized at optimality, and so lose the theoretical guarantees of FM. We demonstrate this below.

Fig. 3 compares FM with $\ell_{\mathrm{MSE}}$ and PH, where $\mathbb{P}_1$ is unit Gaussian and $\mathbb{P}_0$ is a mixture of Gaussians. As shown in Fig. 3a, the two models learn distinct vector fields, so PH indeed induces different ODE trajectories. While the model trained with $\ell_{\mathrm{MSE}}$ translates between $\mathbb{P}_0$ and $\mathbb{P}_1$ accurately, the model trained with PH generates incorrect distributions, *e.g.*, the model density is not isotropic at $t = 1$, and modes are biased at $t = 0$. So, instead of relying on empirical arguments (*e.g.*, Section 4 in Lee et al. (2024)) to justify heuristic losses, we show that a proper choice of the invertible linear map $\phi$ in Eq. (12) can still offer non-trivial performance gains while adhering to FM theory.

**Our improvement.** Previous works have observed high-frequency features are crucial to diffusion-based modeling of image datasets (Kadkhodaie et al., 2024; Zhang & Hooi, 2023; Yang et al., 2023).

| Loss | CIFAR10 | AFHQv2 | FFHQ |
|------|---------|--------|------|
| $\ell_{\mathrm{MSE}}$ | 2.62 | 2.69 | 3.77 |
| PH | 2.59 | 2.71 | 3.75 |
| LPIPS | 2.81 | 2.65 | 4.02 |
| LPIPS+PH | 2.63 | 2.72 | 3.79 |
| $\ell_{\boldsymbol{I}+0.1\,\mathrm{HPF}}$ | 2.63 | 2.62 | 3.76 |
| $\ell_{\boldsymbol{I}+10\,\mathrm{HPF}}$ | **2.58** | **2.55** | **3.69** |
| $\ell_{\boldsymbol{I}+1000\,\mathrm{HPF}}$ | 3.20 | 2.79 | 4.43 |

Table 4: Comparison of various $\ell$ combined with our $w(\boldsymbol{x}_t, t)$ and $\mathbb{T}$.

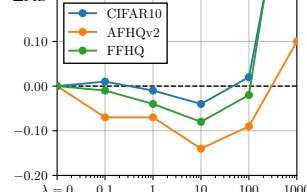

Figure 4: $\lambda$ ablation.

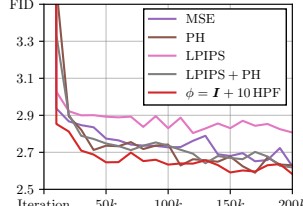

Figure 5: CIFAR10 training.

Moreover, since we initialize $\boldsymbol{D}_\theta$ with a pre-trained DM, we assert that the model already has a good representation of low-frequency visual features. Hence, to accelerate the learning of high-frequency features, we propose calculating the difference of denoiser output $\boldsymbol{D}_\theta(\boldsymbol{x}_t, t)$ and clean data $\boldsymbol{x}_0$ after passing them through a high-pass filter (HPF) using the linear map, *i.e.*, use $\ell_\phi$ in Eq. (12) with

$$\phi = \boldsymbol{I} + \lambda \cdot \mathrm{HPF} \qquad (16)$$

where $\lambda > 0$ controls the emphasis on high-frequency features. The identity matrix in Eq. (16) is necessary to ensure that $\phi$ is invertible, so Eq. (1) is minimized at optimality per Prop. 1. We also remark that using $\ell_\phi$ can be interpreted as preconditioning the gradient. Specifically, since

$$\nabla_{\boldsymbol{D}_\theta} \ell_\phi(\boldsymbol{x}_0, \boldsymbol{D}_\theta(\boldsymbol{x}_t, t)) = \phi^\top \phi\{\nabla_{\boldsymbol{D}_\theta} \ell_{\mathrm{MSE}}(\boldsymbol{x}_0, \boldsymbol{D}_\theta(\boldsymbol{x}_t, t))\}, \qquad (17)$$

using $\ell_\phi$ in place of $\ell_{\mathrm{MSE}}$ is equivalent to scaling the original FM loss gradient along the eigenvectors of $\phi^\top \phi$ by the corresponding eigenvalues. For instance, $\ell_\phi$ with Eq. (16) amplifies gradient magnitudes along high-frequency features by $(\lambda + 1)$, and leaves gradient magnitudes along low-frequency features unchanged. This perspective provides another justification for using $\ell_\phi$, since using an appropriate preconditioning matrix can accelerate convergence (Kingma & Ba, 2015).

**Ablations.** Tab. 4 compares our loss $\ell_{\boldsymbol{I}+\lambda\,\mathrm{HPF}}$ for $\lambda \in \{0.1, 10, 1000\}$ with $\ell_{\mathrm{MSE}}$ and the heuristic losses. If $\lambda$ is too small, $\ell_{\boldsymbol{I}+\lambda\,\mathrm{HPF}}$ has little improvement compared to $\ell_{\mathrm{MSE}}$, whereas if $\lambda$ is too large, $\phi$ becomes nearly singular, leading to a severe drop in the FID. Our loss with $\lambda = 10$ provides consistent improvement over $\ell_{\mathrm{MSE}}$, doing even better than PH and LPIPS. Indeed, in Fig. 4 which visualizes FID change w.r.t. $\ell_{\mathrm{MSE}}$ for various values of $\lambda$, we observe $\lambda = 10$ provides the optimal performance across all datasets. Morever, CIFAR10 learning curves in Fig. 5 verify that $\ell_{\boldsymbol{I}+10\,\mathrm{HPF}}$ enjoys fast convergence compared to all other losses.

## 3.3 IMPROVING LEARNING

### 3.3.1 MODEL DROPOUT

**Previous practice.** Similar to *simple diffusion* (Hoogeboom et al., 2023), we find dropout to be highly impactful. There is little study on the impact of dropout in denoiser UNets for ReFlow. Dropout rates in ReFlow denoiser UNets are usually set to $0.15$ (Liu et al., 2022; Zhu et al., 2024), or equal to the dropout rates of DMs that are used to initialize ReFlow denoisers (Lee et al., 2024). For the EDM networks, dropout rates are $0.13$ on CIFAR10, $0.25$ on AFHQv2, and $0.05$ on FFHQ.

**Our improvement.** We observe that learning a straight ODE is a harder task than learning the diffusion probability flow ODE. For instance, at $t = 1$, the optimal DM denoiser only needs to predict the data mean $\mathbb{E}_{\boldsymbol{x}_0 \sim \mathbb{P}_0}[\boldsymbol{x}_0]$ for any input $\boldsymbol{x}_1 \sim \mathbb{P}_1$, but a denoiser for a perfectly straight ODE has to directly map $\mathbb{P}_1$ samples to $\mathbb{P}_0$ samples. This means we need a larger Lipschitz constant for the ReFlow denoiser (Salmona et al., 2022) (see Appendix G for further discussion), so we use smaller dropout rates during ReFlow training in favor of larger UNet capacity over regularization.

**Ablations.** To verify that smaller dropout rates are beneficial, we return to the baseline training setting (Tab. 1), and run a grid search over dropout probability $p \in [0, 0.15]$. In Fig. 6, which shows FID change w.r.t. baseline $p = 0.15$, we find that smaller $p$ is always beneficial. In fact, optimal $p$ are even smaller than those used to train EDM denoisers, despite using the same architecture (Tab. 5). FIDs after applying optimal dropout to baseline are written in row BSL+OP of Tab. 6. We also observe in row DYN+OP that optimal dropout rates can be combined with improved dynamics to further enhance performance without additional grid search over $p$.

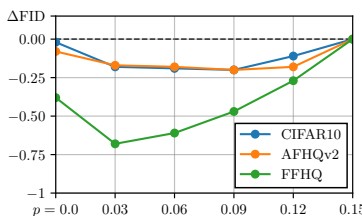

Figure 6: Dropout $p$ ablation.

| | CIFAR10 | AFHQv2 | FFHQ |
|---|---|---|---|
| RF $p =$ | 0.15 | 0.15 | 0.15 |
| EDM $p =$ | 0.13 | 0.25 | 0.05 |
| Ours $p =$ | 0.09 | 0.09 | 0.03 |

Table 5: Dropout $p$ in each setting.

| | CIFAR10 | AFHQv2 | FFHQ |
|---|---|---|---|
| **Baseline (BSL)** | 2.83 | 2.87 | 4.28 |
| **Dynamics (DYN)** | | | |
| BSL $+ w(\boldsymbol{x}_t, t)$ | $2.61_{\triangledown 0.22}$ | $2.74_{\triangledown 0.13}$ | $3.83_{\triangledown 0.45}$ |
| BSL $+ w(\boldsymbol{x}_t, t) + \mathbb{T}$ | $2.62_{\triangledown 0.21}$ | $2.69_{\triangledown 0.18}$ | $3.77_{\triangledown 0.51}$ |
| BSL $+ w(\boldsymbol{x}_t, t) + \mathbb{T} + \ell_\phi$ | $2.58_{\triangledown 0.25}$ | $2.55_{\triangledown 0.32}$ | $3.69_{\triangledown 0.59}$ |
| **Learning (LRN)** | | | |
| BSL $+$ Optimal $p$ (OP) | $2.63_{\triangledown 0.20}$ | $2.67_{\triangledown 0.20}$ | $3.60_{\triangledown 0.68}$ |
| BSL $+$ OP $+$ Forward | $2.57_{\triangledown 0.26}$ | $2.63_{\triangledown 0.24}$ | $3.60_{\triangledown 0.68}$ |
| BSL $+$ OP $+$ Projected | $2.57_{\triangledown 0.26}$ | $2.62_{\triangledown 0.25}$ | $3.58_{\triangledown 0.70}$ |
| **DYN & LRN** | | | |
| DYN $+$ OP | $\underline{2.43}_{\triangledown 0.40}$ | $2.53_{\triangledown 0.34}$ | $3.17_{\triangledown 1.11}$ |
| DYN $+$ OP $+$ Forward | $\mathbf{2.38}_{\triangledown 0.45}$ | $\mathbf{2.44}_{\triangledown 0.43}$ | $\underline{3.14}_{\triangledown 1.14}$ |
| DYN $+$ OP $+$ Projected | $\mathbf{2.38}_{\triangledown 0.45}$ | $\underline{2.47}_{\triangledown 0.40}$ | $\mathbf{3.13}_{\triangledown 1.15}$ |

Table 6: Summary of our training improvements. Subscripts denote FID improvement w.r.t. baseline. Evaluated with sigmoid discretization (see Append. F.1).

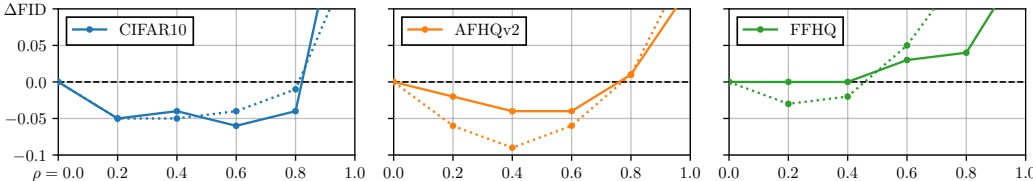

Figure 7: $\rho$ ablation. Solid and dotted lines show results w/o and with improved dynamics, resp.

### 3.3.2 TRAINING COUPLING

**Previous practice.** A common practice is to generate a large number of pairs from $\mathbb{Q}_{01}^1$ by solving the diffusion probability flow ODE Eq. (6) backwards, *i.e.*, from noise to data, and use the generated set as an empirical approximation of $\mathbb{Q}_{01}^1$ throughout training (Liu et al., 2022; Lee et al., 2024; Zhu et al., 2024; Liu et al., 2024). However, the set of generated $\boldsymbol{x}_0$ is only an approximation of the true marginal $\mathbb{P}_0$, so naively training with generated data will accumulate error on the marginal at $t = 0$, as discussed by Alemohammad et al. (2024) (see Appendix G for further discussion).

**Our improvement – *forward pairs*.** To mitigate error accumulation at $t = 0$, we incorporate pairs generated by solving the diffusion probability flow ODE forwards, starting from data, coined forward pairs. We assert forward pairs can be helpful, as $\boldsymbol{x}_0$ are exactly data points.

To use forward pairs, we first invert the training sets for each dataset, which yields additional $50k$ pairs for CIFAR10, $13.5k$ pairs for AFHQv2, and $70k$ pairs for FFHQ. Due to the small number of forward pairs, we use them in combination with backward pairs, and to prevent forward pairs from being ignored due to the large number of backward pairs, we sample forward pairs with probability $\rho$ and backward pairs with probability $1 - \rho$ at each step of the optimization.

**Our improvement – *projected pairs*.** We also propose projecting the coupling $\mathbb{Q}_{01}^1$ to $\Pi(\mathbb{P}_0, \mathbb{P}_1)$, the set of joint distributions with marginals $\mathbb{P}_0$ and $\mathbb{P}_1$, by solving the optimization problem

$$\widehat{\mathbb{Q}}_{01}^1 = \arg \min_{\Gamma_{01}} W_p(\Gamma_{01}, \mathbb{Q}_{01}^1) \quad s.t. \quad \Gamma_{01} \in \Pi(\mathbb{P}_0, \mathbb{P}_1) \tag{18}$$

where $W_p$ is the $p$-Wasserstein distance (Villani, 2009), and using the projected coupling $\widehat{\mathbb{Q}}_{01}^1$ in place of the original during training. Intuitively, this procedure can be understood as fine-tuning the generated marginals to adhere to the true marginals without losing the coupling information in $\mathbb{Q}_{01}^1$. We do not mix projected pairs with any other pairs. The full procedure is described in Appendix D.

**Ablations.** In Fig. 7, we see that it is always beneficial to use forward pairs, as long is $\rho$ is not too high, *e.g.*, $\rho \leq 0.5$. Otherwise, the model starts overfitting to the forward pairs. Interestingly, on FFHQ, using forward pairs without improved training dynamics has no improvement in the FID, implying that improved dynamics may be necessary to make the best out of the rich information contained in the forward pairs. In rows BSL+OP+Projected and DYN+OP+Projected of Tab. 6, we observe that projected pairs also offer improvements in the FID score across all three datasets.

|        | CIFAR10 | AFHQv2 | FFHQ |
|--------|---------|--------|------|
| Unif.  | $2.36_{\nabla 0.47}$ | $2.34_{\nabla 0.53}$ | $2.97_{\nabla 1.31}$ |
| EDM    | $2.80_{\nabla 0.03}$ | $3.61_{\triangle 0.74}$ | $6.78_{\triangle 2.50}$ |
| **Ours** | | | |
| $\kappa = 10$ | $\underline{2.31}_{\nabla 0.52}$ | $\underline{2.31}_{\nabla 0.56}$ | $\underline{2.87}_{\nabla 1.41}$ |
| $\kappa = 20$ | $\mathbf{2.23}_{\nabla 0.60}$ | $\mathbf{2.30}_{\nabla 0.57}$ | $\mathbf{2.84}_{\nabla 1.44}$ |
| $\kappa = 30$ | $2.45_{\nabla 0.38}$ | $2.78_{\nabla 0.09}$ | $3.32_{\nabla 0.96}$ |

Table 7: Various discretizations applied to our best models and DPM-Solver with $r = 0.4$.

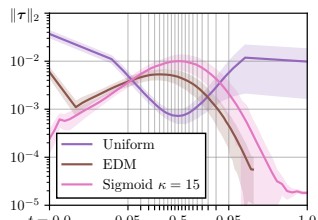

Figure 8: Truncation error.

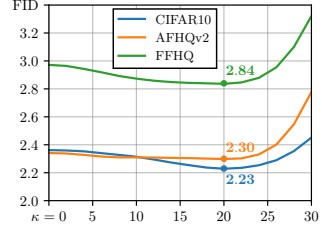

Figure 9: $\kappa$ ablation.

### 3.4 IMPROVING INFERENCE

**Previous practice.** To generate data after ReFlow, previous works often use an uniform discretization $\{t_i = i/N : i = 0, \dots, N\}$ of $[0, 1]$ along with the Euler or Heun to integrate Eq. (3) from $t = 1$ to $0$ (Liu et al., 2022; 2024; Lee et al., 2024; Zhu et al., 2024).

**Our improvement.** As ReFlow ODE converges to a straight ODE, we assert that high-curvature regions in ODE paths now occur near $t \in \{0, 1\}$. While the previously proposed EDM schedule

$$t_0 = 0, \quad t_i = \frac{\sigma_i}{\sigma_{i+1}} \ \text{ where } \ \sigma_i = (\sigma_{\min}^{1/d} + \tfrac{i}{N}(\sigma_{\max}^{1/d} - \sigma_{\min}^{1/d}))^d$$

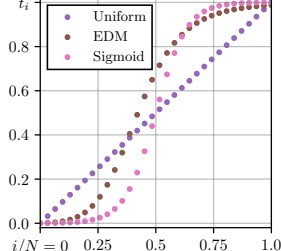

Figure 10: Discretizations.

for solving diffusion probability flow ODEs emphasizes $t \in \{0, 1\}$, we note that it does not perform better than the uniform discretization, as shown in Tab. 7. Similar to Lin et al. (2024), we speculate that this is because $t_N < 1$. Specifically, $\boldsymbol{v}_\theta(\boldsymbol{x}_1, t_N) \neq \boldsymbol{v}_\theta(\boldsymbol{x}_1, 1)$ since $t_N \neq 1$, but the integration of the ODE is done with $\boldsymbol{v}_\theta(\boldsymbol{x}_1, t_N)$ in place of $\boldsymbol{v}_\theta(\boldsymbol{x}_1, 1)$, leading to erroneous ODE trajectories.

Instead of tuning $\{\sigma_{\min}, \sigma_{\max}, d\}$ to address this problem, we propose a simple sigmoid schedule

$$\{t_i = (\text{sig}(\kappa(\tfrac{i}{N} - 0.5)) - \text{sig}(-\tfrac{\kappa}{2}))/(\text{sig}(\tfrac{\kappa}{2}) - \text{sig}(-\tfrac{\kappa}{2})) : i = 0, \dots, N\} \tag{19}$$

with one parameter $\kappa$ which controls the concentration of $t_i$ at $t \in \{0, 1\}$. Here, sig is the sigmoid function. As $\kappa \to 0$, $\{t_i\}$ converges to the uniform discretization, and as $\kappa \to \infty$, all $t_i$ with $i < N/2$ will converge to 0, and all $t_i$ with $i > N/2$ will converge to 1.

To solve the ODE Eq. (3), we consider DPM-Solver (Lu et al., 2022) with the update rule

$$\boldsymbol{x}_{t_i} \leftarrow \boldsymbol{x}_{t_{i+1}} + (t_i - t_{i+1})(\tfrac{1}{2r}\boldsymbol{v}_\theta(\boldsymbol{x}_{s_{i+1}}, s_{i+1}) + (1 - \tfrac{1}{2r})\boldsymbol{v}_\theta(\boldsymbol{x}_{t_{i+1}}, t_{i+1})) \tag{20}$$

where $s_{i+1} = t_i^r t_{i+1}^{1-r}$ and $r \in (0, 1]$. We recover the second order Heun update (the baseline solver) with $r = 1$, but we assert that we can obtain better performance by tuning $r$.

**Ablations.** In Tab. 7, we display results for solving Eq. (3) with various discretizations and DPM-Solver with $r = 0.4$. First, row $\kappa = 10$ shows that we can indeed mitigate the timestep mismatch problem in the EDM schedule. It also shows we can gain improvements by using $r < 1$ (in the baseline setting, we use the sigmoid schedule with $\kappa = 10$ and Heun). See Appendix F.2 for a full ablation over $r$. Second, rows $\kappa = 20, 30$ tells us we can get even better results by increasing sharpness, but too large $\kappa$ hurts performance.

To investigate the performance difference between discretizations, we visualize local truncation error $\|\boldsymbol{\tau}\|_2$ in Fig. 8, where $\boldsymbol{\tau}$ given time-step $t_i$ and $\boldsymbol{x}_{t_{i+1}} \sim \mathbb{Q}_{t_{i+1}}$ is defined as

$$\boldsymbol{\tau} = (\boldsymbol{x}_{t_{i+1}} + (t_i - t_{i+1})\boldsymbol{v}_\theta(\boldsymbol{x}_{t_{i+1}}, t_{i+1})) - \text{solve}(\boldsymbol{x}_{t_{i+1}}, \boldsymbol{D}_\theta, t_{i+1}, t_i).$$

We first note that the uniform distribution incurs large error near $t \in \{0, 1\}$. This highlights that we indeed must place more points near those $t$ in order to control discretization error. While the EDM schedule has less error at those regions, because $t_N \neq 1$, the mismatch between the initial state $\boldsymbol{x}_1$ and time $t_N$ does not ensure the ODE is solved properly. Finally, we see that our schedule is able to control the error at the extremes. While the error for our schedule increases near $t \approx 0.5$, Fig. 9 tells us we can sacrifice accuracy at intermediate $t$ to prioritize perceptual quality by choosing a large $\kappa$.

| Method | CIFAR10 | | | AFHQv2 | | | FFHQ | | | ImageNet (cond.) | | | Reference |
|---|---|---|---|---|---|---|---|---|---|---|---|---|---|
| | NFE | FID | STN | NFE | FID | STN | NFE | FID | STN | NFE | FID | STN | |
| **DM ODE** | | | | | | | | | | | | | |
| EDM | 35 | 1.97 | 14.19 | 79 | 1.96 | 28.41 | 79 | 2.39 | 27.15 | 79 | 2.30 | 26.76 | (Karras et al., 2022) |
| | 9 | 37.91 | – | 9 | 28.03 | – | 9 | 56.84 | – | 9 | 35.46 | – | ——"—— |
| DPM-Solver | 9 | 4.98 | – | – | – | – | 9 | 9.26 | – | 9 | 6.64 | – | (Lu et al., 2022) |
| AMED-Solver | 9 | 2.63 | – | – | – | – | 9 | 4.24 | – | 9 | 5.60 | – | (Zhou et al., 2024) |
| **FM ODE** | | | | | | | | | | | | | |
| MinCurv | 9 | 8.76 | 5.87 | 9 | 13.63 | 10.45 | 9 | 10.44 | 10.49 | – | – | – | (Lee et al., 2023) |
| FM-OT | 142 | 6.35 | – | – | – | – | – | – | – | 138 | 14.45 | – | (Lipman et al., 2023) |
| OT-CFM | 100 | 4.44 | – | – | – | – | – | – | – | – | – | – | (Tong et al., 2023) |
| MOT-50 | – | – | – | – | – | – | – | – | – | 132 | 11.82 | – | (Pooladian et al., 2023) |
| FM* | 100 | 2.96 | 10.73 | 100 | 2.73 | 16.20 | 100 | 3.30 | 16.71 | – | – | – | Baseline |
| MOT-512* | 100 | 3.29 | 8.77 | 100 | 5.53 | 13.45 | 100 | 4.69 | 14.29 | – | – | – | ——"—— |
| MOT-1024* | 100 | 3.18 | 8.59 | 100 | 5.83 | 13.45 | 100 | 4.84 | 14.07 | – | – | – | ——"—— |
| MOT-4096* | 100 | 3.16 | 8.34 | 100 | 6.18 | 12.68 | 100 | 4.92 | 13.47 | – | – | – | ——"—— |
| ReFlow | 110 | 3.36 | – | – | – | – | – | – | – | – | – | – | (Liu et al., 2022) |
| Simple ReFlow* | 9 | 2.23 | 1.64 | 9 | 2.30 | 3.30 | 9 | 2.84 | 2.87 | 9 | 3.49 | 2.72 | Ours |
| + Guidance* | 9 | 1.98 | 2.49 | 9 | 1.91 | 5.60 | 9 | 2.67 | 3.24 | 9 | 1.74 | 3.92 | ——"—— |

Table 8: Comparison of neural ODE methods. MOT-$b$ is minibatch OT with minibatch size $b$, and MinCurv is curvature minimizing flow. We report FID and straightness (STN): $S(\boldsymbol{v}_\theta) \coloneqq \int_0^1 \mathbb{E}\left[\|(\boldsymbol{x}_1 - \boldsymbol{x}_0) - \boldsymbol{v}_\theta(\boldsymbol{x}_t, t)\|_2\right] \, \mathrm{d}t$. Star * next to a method denotes our training results.

## 4 APPLICATIONS

**Comparison to other fast flow methods.** Our approach significantly outperforms other ODE approaches, *e.g.*, minibatch-OT FM (Pooladian et al., 2023; Tong et al., 2023) and curvature minimization (Lee et al., 2023), see see Tab. 8. Where possible, we report straightness (Liu et al., 2022), which quantifies how an ODE trajectory deviates from a straight line between its initial and terminal points (see Tab. 8 and Eq. (21) and further details in Appendix D). We attribute the inferior baseline performance due to bias in minibatch OT, and discuss this and other pitfalls in Appendix A.

**Improving perceptual quality via guidance.** DMs often use guidance such as classifier-free guidance (CFG) (Ho & Salimans, 2022) or autoguidance (AG) (Karras et al., 2024) to enhance the perceptual quality of samples. As observed by Liu et al. (2024), conditional ReFlow models can also be combined with CFG. While it is unclear what effect guidance has on the marginals of ReFlow models, we also apply AG / CFG to our best unconditional / conditional models, since perceptual quality may be of interest for certain downstream tasks. We already achieve state-of-the-art results for fast ODE-based generation, but we obtain even lower FID scores with guidance, as shown in the last row of Tab. 8. See Appendix F.2 for a full ablation over guidance strength.

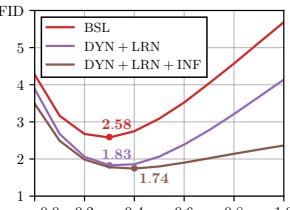

Figure 11: ImageNet-64 FID at 9 NFEs without and with our improvements.

**Class-conditional ImageNet-64.** We verify the scalability of our training dynamics (DYN), learning (LRN), and inference (INF) choices on ReFlow with the class-conditional ImageNet-64 EDM model. We use $8M$ backward, $4M$ forward pairs, and $\rho = 0.2$. Fig. 11 at CFG scale $w = 0$, *i.e.*, no guidance, confirms that our techniques are effective. DYN+LRN improves BSL FID from $4.27$ to $3.91$, and INF further improves the FID to $3.49$, improving on prior state-of-the-art fast flow methods. With CFG $w = 0.4$, our DYN+LRN+INF model achieves an even better FID score of $1.74$. Also, our techniques consistently improve CFG FIDs, implying that they offer orthogonal benefits.

## 5 CONCLUSION

We decompose the design space of ReFlow into training dynamics, learning, and inference. Within each group, we examine prior practices and their potential pitfalls. We propose seven improved choices for loss weight, time distribution, loss function, model dropout, training data, ODE discretization and solver. We verify the robustness of our techniques on CIFAR10, AFHQv2, and FFHQ, and their scalability on ImageNet-64. Our techniques yield SoTA results among fast neural ODE methods, without latent-encoders, perceptual losses, or premetrics. In terms of FID score, weight and dropout contributed most. However, in terms of novelty, we believe training data improvements and the generalized loss function are our largest contributions.

ACKNOWLEDGMENTS

This work was supported by the National Research Foundation of Korea under Grant RS-2024-00336454 and by the Institute of Information & communications Technology Planning & Evaluation (IITP) grant funded by the Korea government (MSIT) (No.2019-0-00075, Artificial Intelligence Graduate School Program (KAIST)).

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

## A  FASTER ODEs VIA COUPLINGS

### A.1  FASTER SAMPLING VIA STRAIGHT PATHS

Generative ODEs with straight trajectories can be solved accurately with substantially fewer velocity evaluations than those with high-curvature trajectories (Stoer & Bulisch, 2002). In fact, probability flow ODEs with perfectly straight trajectories can translate one distribution to another with a single Euler step. For an ODE with velocity $v_\theta$, we can quantify its straightness as (Liu et al., 2022):

$$S(v_\theta) := \int_0^1 \mathbb{E}\left[\|(x_1 - x_0) - v_\theta(x_t, t)\|_2\right] \, dt \tag{21}$$

where the expectation is over ODE trajectories $\{x_t : t \in [0, 1]\}$ generated by $v_\theta$. An ODE with zero straightness has linear trajectories, which means it can translate initial points to terminal points with a single function evaluation.

One approach to encourage straight paths is to learn ODEs which minimize trajectory curvature (Lee et al., 2023) by parameterizing the coupling with a neural network which takes as input some image and outputs a sample such the distribution of these samples is close to Gaussian. Another approach is based on connections to optimal transport.

### A.2  CONNECTION TO OPTIMAL TRANSPORT

For any convex ground cost, the solution to the dynamic optimal transport on continuous support will be straight trajectories, see e.g. Liu (2022). In addition, training a flow matching model on samples from an optimal transport coupling will preserve the coupling, providing the vector field of the flow matching model is conservative.

As an aside, more generally performing bridge matching (Peluchetti, 2021; Shi et al., 2023) on an entropically-regularized optimal coupling will preserve the coupling, though the trajectory will be given by an SDE, and hence no longer straight. In the limit as the entropic regularization term tends to zero, then this recovers the non-regularized optimal coupling with squared Euclidean ground cost (Shi et al., 2023; Peluchetti, 2023).

This motivates learning an optimal transport (OT) coupling and then performing bridge / flow matching on this coupling. There are two dominant ways to do this, either ReFlow (also known as Iterative Markovian fitting) (Liu et al., 2022; Liu, 2022; Lee et al., 2024; Shi et al., 2023) or approximating the coupling using mini-batches (Tong et al., 2023; Pooladian et al., 2023).

#### A.2.1  MINI-BATCH OPTIMAL TRANSPORT FLOW MATCHING

We first make a distinction between the loss batch size $b_{\text{loss}}$ and coupling batch size $b_{\text{coupling}}$ where $b_{\text{coupling}} \geq b_{\text{loss}}$. The loss batch size, $b_{\text{loss}}$, is the number of input pairs $(x_0, x_1)$ used per training iteration in the flow matching loss Eq. (1), whereas the coupling batch size, $b_{\text{coupling}}$, refers to the number of independently sampled pairs used as input into the mini-batch OT solver.

The procedure for obtaining mini-batch couplings is to first independently sample $b_{\text{coupling}}$ items denoted $(y_i)_{i=1}^{b_{\text{coupling}}}$ $(x_i)_{i=1}^{b_{\text{coupling}}}$ from both marginal distributions $x_i \sim \mathbb{P}_0$ and $y_i \sim \mathbb{P}_1$. The next step is to run a mini-batch OT solver to obtain a coupling matrix $P = (p_{i,j})_{i=1,j=1}^{b_{\text{coupling}}}$ such that $\sum_{i,j} p_{i,j} = 1, \sum_i p_{i,j} = 1/b_{\text{coupling}}, \sum_j p_{i,j} = 1/b_{\text{coupling}}$.

The final step is to sub-sample the coupling batch of size $b_{\text{coupling}}$ to obtain $b_{\text{loss}}$ aligned pairs $(\tilde{x}_i, \tilde{y}_i)_{i=1}^{b_{\text{loss}}} \sim P$, which are then fed into loss Eq. (1) and corresponding standard gradient based optimization procedure.

**Mini-batch bias**. In stochastic gradient descent, for example, losses computed on uniformly sampled batches are unbiased with respect to the measures the batches were sampled from. This is not true for mini-batch OT couplings with respect to the true OT coupling between marginals (Bellemare et al., 2017). Indeed, marginal preservation within mini-batches may force points in each minibatchto be mapped together, such points may not be mapped, or have very low probability of being mapped, in the true OT coupling. Asymptotically the mini-batch couplings should converge to the true OT coupling solvers the mini-batch size increases. Unfortunately, this is not practically feasible with discrete OT solvers for large datasets or indeed for measures with continuous support. The mini-batch and true OT couplings would also be the same for infinite regularization, indeed the couplings would both be independent couplings and so not very informative.

**Subsampling**. Computing OT couplings for large batch sizes is not typically possible using the Hungarian algorithm (Kuhn, 1955) due to the cubic time complexity. However, entropic approximations from Sinkhorn (Sinkhorn, 1964) is of only quadratic complexity and can be implemented on modern GPU-accelerators (Cuturi, 2013; Cuturi et al., 2022), hence enables fast computation of discrete entropic OT for batches in excess of $100,000$ points.

Prior works (Tong et al., 2023; Pooladian et al., 2023) set $b_{\text{loss}} = b_{\text{coupling}}$ and do not subsample. This is problematic as mini-batch OT is only justified as being close to optimal in the asymptotically large batch regime. Although we can compute the coupling for large batch sizes, the optimization setup for training the neural network via FM is limited by hardware memory and so it becomes infeasible to set $b_{\text{loss}} = b_{\text{coupling}}$ for large batch size. Prior works therefore use small coupling batch size.

Subsampling should still preserve marginal distributions. We observe in Tab. 8 that the straightness of the generative trajectories increases as batch size grows, as expected, however generative performance in terms of FID gets increasingly worse compared to regular flow matching. This is a surprising empirical result that warrants further investigation.

### A.2.2 ReFlow and Iterative Markovian Fitting

ReFlow (Liu et al., 2022; Liu, 2022; Lee et al., 2024) and more generally Iterative Markovian Fitting (Shi et al., 2023) are procedures which iteratively refine the coupling between marginals. We shall focus on ReFlow for brevity. ReFlow first takes an independent coupling, then involves training a flow between samples from that coupling, known as a Markovian projection. Simulating from this trained flow is then used to define an updated coupling. This process is repeated between updating a flow and coupling until convergence. It has been shown that this process iteratively reduces the transport cost for any convex ground cost, and hence straightens the paths between coupling whilst retaining the correct marginals.

Note that ReFlow results in a coupling which is slightly stronger than optimal transport. OT aims to minimize the transport cost for a specific ground cost function, whereas ReFlow reduces transport cost for all convex costs. ReFlow can be limited to specific convex ground cost by ensuring the vector field takes a specific conservative form (Liu, 2022).

## B Fast Sampling via Higher Order Solvers

One can use higher-order solvers which utilize higher order differentials of the ODE velocity to take large integration steps or reduce truncation error (Karras et al., 2022; Dockhorn et al., 2022; Lu et al., 2022; Zhang & Chen, 2023). While this approach is generally training-free, recent works (Zhou et al., 2024; Kim et al., 2024b) have incorporated trainable components which minimize truncation error to further accelerate sampling.

## C Distillation and Consistency Models

The goal of distillation within the field of diffusion models is typically to compress multiple steps along a probability flow ODE of a teacher diffusion model into a fewer steps steps of a student model. We refer to this as discrete-time distillation (DTD). Representative methods are progressive distillation (Salimans & Ho, 2022), and consistency distillation (Song et al., 2023; Song & Dhariwal, 2024; Kim et al., 2024a; Geng et al., 2024). While distillation and ReFlow are similar in the aspect

|                     | CIFAR10 | AFHQv2 | FFHQ  | ImageNet-64 |
|---------------------|---------|--------|-------|-------------|
| Iterations          | $200k$  | $200k$ | $200k$| $500k$      |
| Minibatch Size      | 512     | 256    | 256   | 1024        |
| Adam LR             | 2e−4    | 2e−4   | 2e−4  | 2e−4        |
| Label dropout       | –       | –      | –     | 0.1         |
| EMA                 | 0.9999  | 0.9999 | 0.9999| 0.9999      |
| Num. Backward Pairs | $1M$    | $1M$   | $1M$  | $8M$        |
| Num. Forward Pairs  | $50k$   | $13.5k$| $70k$ | $4M$        |

Table 9: Training hyper-parameters.

that they train a new model using the outputs a teacher diffusion model, we emphasize that they are, in fact, complementary approaches, and can benefit from one another. We discussion this point in more detail in the following section.

### C.1 ReFlow vs. Distillation

We remark that faster ODEs have several practical benefits over discrete-time distillation alone. Since translation along an ODE is a bijective map, we can achieve fast inversion and likelihood evaluation by integrating the ODE backwards starting from data.

Fast ODEs can be combined with discrete-time distillation. For instance, Lee et al. (2023); Liu et al. (2024); Zhu et al. (2024) have observed it is substantially easier to distill ODE models with straight trajectories. One may also use any ODE solver with a continuous time ODE, and there may be some benefit using adaptive solvers. Lee et al. (2023; 2024) also report combining RF models with higher-order solvers improves the trade-off between generation speed and quality.

## D  Experiment Settings

### D.1 Training and Evaluation

To evaluate a training setting, we initialize ReFlow denoisers with pre-trained EDM (Karras et al., 2022) denoisers, and optimize Eq. (13) with $\mathbb{Q}_{01} = \mathbb{Q}_{01}^1$ of Eq. (7). Specific optimization hyper-parameters are reported in Tab. 9. We sample backward and forward pairs from $\mathbb{Q}_{01}^1$ by solving Eq. (6) with EDM models and use them throughout training. Specifically, we use the EDM discretization with the Heun solver (Ascher & Petzold, 1998). We use sampling budgets of 35 NFEs for CIFAR10 and 79 NFEs for AFHQv2, FFHQ, and ImageNet. FIDs of backward training samples are reported in the first row of Tab. 8.

We measure the generative performance of the optimized model by computing the FID (Heusel et al., 2017) between $50k$ generated images and all available dataset images. Inception statistics are computed using the pre-trained Inception-v3 model (Karras et al., 2023a). Samples are generated by solving Eq. (3) with the Heun solver with 9 NFEs, and we report the minimum FID score out of three random generation trials, as done by Karras et al. (2022). For reasons described in Appendix F.1, we use the sigmoid discretization instead of the baseline uniform discretization.

#### D.1.1 Best Settings

Here, we report hyper-parameters used to produce results for our best models in Table 8.

**CIFAR10.** High-pass filter $\lambda = 10$, dropout probability 0.09, forward pairs with mixing ratio $\rho = 0.4$, sigmoid discretization with $\kappa = 20$, DPM-solver $r = 0.4$, AutoGuidance scale $w = 0.6$.

**AFHQv2.** High-pass filter $\lambda = 10$, dropout probability 0.09, forward pairs with mixing ratio $\rho = 0.4$, sigmoid discretization with $\kappa = 20$, DPM-solver $r = 0.4$, AutoGuidance scale $w = 1.0$.

**FFHQ.** High-pass filter $\lambda = 10$, dropout probability 0.03, forward pairs with mixing ratio $\rho = 0.2$, sigmoid discretization with $\kappa = 20$, DPM-solver $r = 0.4$, AutoGuidance scale $w = 0.3$.

**ImageNet-64.** High-pass filter $\lambda = 10$, dropout probability 0.05, forward pairs with mixing ratio $\rho = 0.2$, sigmoid discretization with $\kappa = 20$, DPM-solver $r = 0.4$, CFG scale $w = 0.4$.

## D.2 FLOW MATCHING BASELINES

We strove to obtain competitive baselines for base and mini-batch OT flow matching methods, and indeed achieved superior performance to comparable implementations from Tong et al. (2023) on the datasets considered.

Firstly, similar to Karras et al. (2022), we formulate flow matching as $x_0$ or *mean*-prediction rather than using regression target $X_0 - X_1$. We parameterize the mean-prediction to be of form $D_\theta(x_t, t) = c_{\text{skip}}(t)x_t + c_{\text{out}}(t)F_\theta(c_{\text{in}}(t), c_\sigma(t))$ where $F_\theta$ is a neural network:

$$\mathbb{E}_{t,\mathbf{X}_t,\mathbf{X}_0}\lambda(t)\|D_\theta(\mathbf{X}_t, t) - \mathbf{X}_0\|^2. \tag{22}$$

The scalar functions $c_\sigma, c_{\text{skip}}, c_{\text{out}}, c_{\text{in}}, \lambda(t)$ are derived according to the reasoning of Karras et al. (2022) in Sec D.2.2. We set $\sigma_{0,T} = 0$ for the independent coupling.

Throughout we use the a similar setup as Karras et al. (2022) but with the flow matching loss and new preconditioning. In particular for AFHQv2, FFHQ and CIFAR10 we use the *SongNet* from Song et al. (2021b) with corresponding hyperparameters from Karras et al. (2022) per dataset.

The time-sampling during training is taken to be uniform and the Euler solver with 100 steps is used for computing FID and straightness metrics, in order to be comparable to other reported baselines from Tong et al. (2023).

### D.2.1 MINI-BATCH FLOW MATCHING

The mini-batch flow matching experiments use the same learning rate, networks, and training objectives as base flow matching. The primary difference is in how the inputs, $\mathbf{X}_t, \mathbf{X}_0$, are sampled.

We follow the procedure outlined in Sec. A.2.1 for sampling mini-batches, using Sinkhorn (Sinkhorn, 1964; Cuturi, 2013) as the mini-batch solver based on the OTT-JAX library (Cuturi et al., 2022). Images were scaled to $[-1, 1]$ as is standard in diffusion models and flattened. The squared Euclidean ground cost was used.

The regularization parameter was set to $\epsilon = 2$, qualitatively this provided a reasonable trade-off between meaningful coupling visually and the time to compute using convergence threshold defaults from Cuturi et al. (2022). The default regularization parameter from Cuturi et al. (2022) did not provide a visually meaningful coupling on large batches, and setting parameter less than $\epsilon < 1$ took over the maximum iteration threshold of $2,000$ iterations to converge, and hence was not feasible for training.

Each Sinkhorn loop took approximately $100 - 200$ Sinkhorn iterations without acceleration techniques, and wall-clock time up to roughly $0.8$s for the largest coupling batch size $8192$. We then ran acceleration techniques including Anderson acceleration (Anderson, 1965) with memory 2, epsilon decay starting from 10, and initializing potentials from prior batches to reduce runtime. This sped up the mini-batch process to $0.4$s per Sinkhorn loop, and convergence of Sinkhorn in approximately $20 - 30$ Sinkhorn iterations.

We ablated the coupling batch size between $512, 1024, 4096, 8192$. The loss batch size was kept constant at $512$ for CIFAR10 and $256$ for AFHQV2 and FFHQ.

**Scope for further improvements:** Unfortunately, $\mathbb{C}ov[X_0, x_T] = \sigma_{0,T}^2$ is not known for mini-batch couplings and hence as in the independent coupling we set $\sigma_{0,T} = 0$ in the preconditioning computation. It is possible that this is sub-optimal and may be estimated in better ways.

Although straightness improves with larger batch size and our implementation achieves better FID scores than prior baselines, mini-batch OT flow matching is still not well understood. It is puzzling as to why performance in terms of FID gets worse compared to base flow matching. This is corroborated in Table 5 of Tong et al. (2023) where FID for CIFAR10 is 3.74 with the mini-batch coupling and 3.64 with independent coupling, however we notice a significant discrepancy at 2.98 FID for the independent coupling and 3.16 for the best mini-batch coupling. We leave further investigations to future work.

### D.2.2 PRECONDITIONING

In the interest of generality, we derive EDM-style preconditioning (Karras et al., 2022) for the more general case of bridge matching / stochastic interpolant (Peluchetti, 2023; 2021; Shi et al., 2023; Albergo et al., 2023) which recovers preconditioning for flow matching for $\gamma_t = 0$,

Let $\mathbf{X}_t = \alpha_t \mathbf{X}_0 + \beta_t \mathbf{X}_T + \gamma_t \epsilon$ where $\epsilon \sim \mathcal{N}(0, \mathbb{I})$. Consider prediction of form $D_\theta(x_t, t) = c_{\text{skip}}(t) x_t + c_{\text{out}}(t) F_\theta(c_{\text{in}}(t), c_\sigma(t))$ and $\lambda(\cdot)$ weighted loss per $Eq.$ (22).

The loss per $Eq.$ (22) may be written:

$$\mathbb{E}_{t, \mathbf{X}_t, \mathbf{X}_0} \lambda(t) c_{\text{out}}(t)^2 \| F_\theta(\mathbf{X}_t, t) - c_{\text{out}}(t)^{-1}(\mathbf{X}_0 - c_{\text{skip}}(t)\mathbf{X}_t) \|^2 \tag{23}$$

**Setting** $\lambda(\cdot)$. In order to uniformly weight the loss per time step, we set $\lambda(t) = c_{\text{out}}(t)^{-2}$ similarly to Karras et al. (2022).

**Setting** $c_{\text{in}}(t)$. We take the strategy of finding $c_{\text{in}}$ such that $\mathbb{V}ar[c_{\text{in}}(t)\mathbf{X}_t] = 1$.

Let $\mathbb{V}ar[\mathbf{X}_0] = \sigma_0^2$, $\mathbb{V}ar[\mathbf{X}_T] = \sigma_T^2$ and $\mathbb{C}ov[\mathbf{X}_0, x_T] = \sigma_{0,T}^2$

$$\mathbb{V}ar[c_{\text{in}}(t)\mathbf{X}_t] = c_{\text{in}}(t)^2 \left[ \alpha_t^2 \sigma_0^2 + \beta_t^2 \sigma_0^2 + 2\alpha_t \beta_t \sigma_{0,T}^2 + \gamma_t^2 \right] = 1 \tag{24}$$

$$c_{\text{in}}(t) = \left[ \alpha_t^2 \sigma_0^2 + \beta_t^2 \sigma_T^2 + 2\alpha_t \beta_t \sigma_{0,T}^2 + \gamma_t^2 \right]^{-\frac{1}{2}} \tag{25}$$

**Setting** $c_{\text{skip}}$ and $c_{\text{out}}$. The prediction target of $D_\theta(x_t, t)$ is $\mathbf{X}_0$, hence the target of network $F_\theta$ is $c_{\text{out}}(t)^{-1}[\mathbf{X}_0 - c_{\text{skip}}(t)x_t]$. We choose $c_{\text{skip}}$ and $c_{\text{out}}$ to ensure regression target has uniform variance i.e. $\mathbb{V}ar\left[ c_{\text{out}}(t)^{-1}[\mathbf{X}_0 - c_{\text{skip}}(t)\mathbf{X}_t] \right] = 1$,

$$\mathbb{V}ar\left[ c_{\text{out}}(t)^{-1}[\mathbf{X}_0 - c_{\text{skip}}(t)\mathbf{X}_t] \right] = 1 \tag{26}$$

$$c_{\text{out}}(t)^2 = \mathbb{V}ar\left[ \mathbf{X}_0 - c_{\text{skip}}(t)\mathbf{X}_t \right] \tag{27}$$

$$c_{\text{out}}(t)^2 = \mathbb{V}ar\left[ (1 - \alpha_t c_{\text{skip}}(t))\mathbf{X}_0 - c_{\text{skip}}(t)(\beta_t \mathbf{X}_T + \gamma_t \epsilon) \right] \tag{28}$$

$$c_{\text{out}}(t)^2 = (1 - \alpha_t c_{\text{skip}}(t))^2 \sigma_0^2 \tag{29}$$

$$- 2\beta_t(1 - \alpha_t c_{\text{skip}}(t))c_{\text{skip}}(t)\sigma_{0,T}^2 \tag{30}$$

$$+ c_{\text{skip}}(t)^2 \beta_t^2 \sigma_T^2 + \gamma_t^2 c_{\text{skip}}(t)^2 \tag{31}$$

Given the fixed relationship between $c_{\text{skip}}$ and $c_{\text{out}}$, we choose $c_{\text{skip}}$ to minimize $c_{\text{out}}$

$$\frac{\mathrm{d}c_{\text{out}}^2}{\mathrm{d}c_{\text{skip}}} = - 2\alpha_t(1 - \alpha_t c_{\text{skip}}(t))\sigma_0^2 \tag{32}$$

$$- 2\beta_t \sigma_{0,T}^2 + 4\alpha_t \beta_t \sigma_{0,T}^2 c_{\text{skip}}(t) \tag{33}$$

$$+ 2c_{\text{skip}}(t)\beta_t^2 \sigma_T^2 + 2\gamma_t^2 c_{\text{skip}}(t) \tag{34}$$

With first order condition $\frac{\mathrm{d}c_{\text{out}}^2}{\mathrm{d}c_{\text{skip}}} = 0$, we obtain:

$$c_{\text{skip}}(t) = \frac{\alpha_t \sigma_0^2 + \beta_t \sigma_{0,T}^2}{\alpha_t^2 \sigma_0^2 + 2\alpha_t \beta_t \sigma_{0,T}^2 + \beta_t^2 \sigma_T^2 + \gamma_t^2}. \tag{35}$$

### D.3 COUPLING PROJECTION

Recall that we propose projecting $\mathbb{Q}_{01}^1$ to $\Pi(\mathbb{P}_0, \mathbb{P}_1)$ at the end of each iteration

$$\widehat{\mathbb{Q}}_{01}^1 := \text{proj}_{\Pi(\mathbb{P}_0, \mathbb{P}_1)}(\mathbb{Q}_{01}^1) \tag{36}$$

and using $\widehat{\mathbb{Q}}_{01}^1$ in place of $\mathbb{Q}_{01}^1$. However, the projection operation is well-defined only if there is a suitable metric on the space under consideration (the space of distributions, in our case). An applicable metric is the $p$-Wasserstein distance $W_p$. Then, projection w.r.t. $W_p$ is defined as

$$\widehat{\mathbb{Q}}_{01}^1 = \arg\min_{\Gamma_{01}} W_p(\Gamma_{01}, \mathbb{Q}_{01}^1) \quad s.t. \quad \Gamma_0 = \mathbb{P}_0, \Gamma_1 = \mathbb{P}_1. \tag{37}$$

Furthermore, we may parameterize

$$d\Gamma_{01}(\boldsymbol{x}_0, \boldsymbol{x}_1) = d\Gamma_{0|1}(\boldsymbol{x}_0|\boldsymbol{x}_1)d\mathbb{P}_1(\boldsymbol{x}_1) \quad or \quad d\mathbb{P}_0(\boldsymbol{x}_0)d\Gamma_{1|0}(\boldsymbol{x}_1|\boldsymbol{x}_0) \tag{38}$$

which means (with the first parameterization), we only have to enforce the marginal constraint

$$\widehat{\mathbb{Q}}_{01}^1 = \arg\min_{\Gamma_{01}} W_p(\Gamma_{01}, \mathbb{Q}_{01}^1) \quad s.t. \quad \Gamma_0 = \mathbb{P}_0, \; d\Gamma_{01} = d\Gamma_{0|1}d\mathbb{P}_1. \tag{39}$$

Noting that

$$\Gamma_0 = \mathbb{P}_0 \iff D(\Gamma_0, \mathbb{P}_0) = 0 \tag{40}$$

for distances or divergences $D$, we can optimize

$$\min_{\Gamma_{01}} D(\Gamma_0, \mathbb{P}_0) + \lambda W_p^p(\Gamma_{01}, \mathbb{Q}_{01}^1) \quad s.t. \quad d\Gamma_{01} = d\Gamma_{0|1}d\mathbb{P}_1 \tag{41}$$

for decreasing values of $\lambda$ and stop when $D(\Gamma_0, \mathbb{P}_0)$ saturates. In practice, we solve

$$\min_{\Gamma_{01}} D(\Gamma_0, \mathbb{P}_0) + \lambda \operatorname{SKD}_p(\Gamma_{01}, \mathbb{Q}_{01}^1) \quad s.t. \quad d\Gamma_{01} = d\Gamma_{0|1}d\mathbb{P}_1 \tag{42}$$

with gradient descent, where SKD stands for Sinkhorn Divergence (Feydy et al., 2019). We approximate $\mathbb{Q}_{01}^1$ as a mixture of diracs using the generated backward pairs, and approximate $D$ using a Generative Adversarial Network (Goodfellow et al., 2014). Since we do not know an appropriate value of $\lambda$, we initialize $\lambda$ from a large value, *e.g.*, $\lambda = 1000$, decay it by a factor of 0.1 every time FID saturates. If decaying $\lambda$ does not offer any more FID improvement, we terminate optimization, and use the optimized $\Gamma_{01}$ as $\widehat{\mathbb{Q}}_{01}^1$. The full optimization procedure is described in Algorithm 2.

---

**Algorithm 1** Coupling projection given $\lambda$

---

1: **Inputs:** $\mathbb{P}_0, \mathbb{Q}_{01}^1, \Gamma_{01}$, batch size $B$, discriminator $D_\psi$, discriminator learning rate $\eta$, coupling learning rate $\gamma$, evaluate FID every $N_{\text{FID}}$ iterations, $\operatorname{SKD}_p$ coefficient $\lambda$
2: Initialize $i \leftarrow 0, \Gamma_{01,best} \leftarrow \Gamma_{01}, \operatorname{FID}_{best} \leftarrow \operatorname{FID}(\Gamma_0, \mathbb{P}_0)$
3: **while** training **do**
4:     Sample $\{\tilde{\boldsymbol{x}}_0^n\}_{n=1}^B \sim \mathbb{P}_0, \{(\boldsymbol{x}_0^n, \boldsymbol{x}_1^n)\}_{n=1}^B \sim \mathbb{Q}_{01}^1, \{(\hat{\boldsymbol{x}}_0^n, \hat{\boldsymbol{x}}_1^n)\}_{n=1}^B \sim \Gamma_{01}$
5:     $\psi \leftarrow \phi + \eta\nabla_\psi\{\sum_n \log D_\psi(\tilde{\boldsymbol{x}}_0^n) + \sum_n \log(1 - D_\psi(\hat{\boldsymbol{x}}_0^n))\}$
6:     $\hat{\boldsymbol{x}}_0^n \leftarrow \hat{\boldsymbol{x}}_0^n - \gamma\nabla_{\hat{\boldsymbol{x}}_0^n}\{\log(1 - D_\psi(\hat{\boldsymbol{x}}_0^n)) + \lambda\operatorname{SKD}_p(\{(\hat{\boldsymbol{x}}_0^n, \hat{\boldsymbol{x}}_1^n)\}_{n=1}^B, \{(\boldsymbol{x}_0^n, \boldsymbol{x}_1^n)\}_{n=1}^B)\}$
7:     $i \leftarrow i + 1$
8:     **if** $i \% N_{\text{FID}} = 0$ **then**
9:         $\operatorname{FID}_{curr} \leftarrow \operatorname{FID}(\Gamma_0, \mathbb{P}_0)$
10:        **if** $\operatorname{FID}_{curr} \geq \operatorname{FID}_{best}$ **then**
11:           **return** $\Gamma_{01,best}, \operatorname{FID}_{best}$
12:        **else**
13:           $(\Gamma_{01,best}, \operatorname{FID}_{best}) \leftarrow (\Gamma_{01}, \operatorname{FID}_{curr})$
14:        **end if**
15:     **end if**
16: **end while**

---

**Algorithm 2** Coupling projection

---

1: **Inputs:** $\mathbb{P}_0, \mathbb{Q}_{01}^1$, batch size $B$, discriminator $D_\psi$, discriminator learning rate $\eta$, coupling learning rate $\gamma$, evaluate FID every $N_{\text{FID}}$ iterations, initial $\lambda$, $\lambda$ decay factor $\rho \in (0, 1)$
2: Initialize $i \leftarrow 0, \Gamma_{01} \leftarrow \mathbb{Q}_{01}^1, \Gamma_{01,best} \leftarrow \mathbb{Q}_{01}^1, \operatorname{FID}_{best} \leftarrow \operatorname{FID}(\Gamma_0, \mathbb{P}_0)$
3: **while** training **do**
4:     $\Gamma_{01}, \operatorname{FID}_{curr} \leftarrow \texttt{ALG1}[\mathbb{P}_0, \mathbb{Q}_{01}^1, \Gamma_{01}, B, D_\psi, \eta, \gamma, N_{\text{FID}}, \lambda]$
5:     **if** $\operatorname{FID}_{curr} \geq \operatorname{FID}_{best}$ **then**
6:        **return** $\Gamma_{01,best}$
7:     **else**
8:        $(\lambda, \Gamma_{01,best}, \operatorname{FID}_{best}) \leftarrow (\rho \cdot \lambda, \Gamma_{01}, \operatorname{FID}_{curr})$
9:     **end if**
10: **end while**

---

# E  PROOFS

## E.1  PROOF OF PROPOSITION 1

**Lemma 1.** *The following statements are equivalent.*

*(a) $\theta$ minimizes $\mathcal{L}_{\mathrm{FM}}(\theta; \mathbb{Q}_{01}, \boldsymbol{x}_t, t)$.*

*(b) $\theta$ minimizes $\mathcal{L}_{\mathrm{GFM}}(\theta; \mathbb{Q}_{01}, \boldsymbol{x}_t, t)$.*

*(c) $\boldsymbol{D}_\theta(\boldsymbol{x}_t, t) = \mathbb{E}_{\boldsymbol{x}_0 \sim \mathbb{Q}_{0|t}(\cdot | \boldsymbol{x}_t)}[\boldsymbol{x}_0]$.*

*Proof.* We first observe that (writing $\boldsymbol{D}_\theta$ in place of $\boldsymbol{D}_\theta(\boldsymbol{x}_t, t)$ for brevity)

$$\nabla_{\boldsymbol{D}_\theta} \mathcal{L}_{\mathrm{GFM}}(\theta; \mathbb{Q}_{01}, \boldsymbol{x}_t, t) = \phi^\top \phi \{\nabla_{\boldsymbol{D}_\theta} \mathcal{L}_{\mathrm{FM}}(\theta; \mathbb{Q}_{01}, \boldsymbol{x}_t, t)\} \tag{43}$$

and since $\phi$ is invertible, $\phi^\top \phi$ is invertible as well, which implies

$$\nabla_{\boldsymbol{D}_\theta} \mathcal{L}_{\mathrm{GFM}}(\theta; \mathbb{Q}_{01}, \boldsymbol{x}_t, t) = \boldsymbol{0} \iff \nabla_{\boldsymbol{D}_\theta} \mathcal{L}_{\mathrm{FM}}(\theta; \mathbb{Q}_{01}, \boldsymbol{x}_t, t) = \boldsymbol{0}. \tag{44}$$

Because both $\mathcal{L}_{\mathrm{GFM}}(\theta; \mathbb{Q}_{01}, \boldsymbol{x}_t, t)$ and $\mathcal{L}_{\mathrm{FM}}(\theta; \mathbb{Q}_{01}, \boldsymbol{x}_t, t)$ are strongly convex w.r.t. $\boldsymbol{D}_\theta$, this means $\theta$ minimizes $\mathcal{L}_{\mathrm{GFM}}(\theta; \mathbb{Q}_{01}, \boldsymbol{x}_t, t)$ iff $\theta$ minimizes $\mathcal{L}_{\mathrm{FM}}(\theta; \mathbb{Q}_{01}, \boldsymbol{x}_t, t)$ iff

$$\boldsymbol{D}_\theta(\boldsymbol{x}_t, t) = \mathbb{E}_{\boldsymbol{x}_0 \sim \mathbb{Q}_{0|t}(\cdot | \boldsymbol{x}_t)}[\boldsymbol{x}_0]. \tag{45}$$

This establishes the equivalence of the three claims. $\square$

**Lemma 2.** *Let $\mu$ be a $\sigma$-finite measure. If $f > g$ on a set $A$ with $\mu(A) > 0$, $\int_A f \, d\mu > \int_A g \, d\mu$.*

*Proof.* By linearity of integrals, we can assume $g = 0$. Since $f > 0$ on $A$, we may express

$$A = \cup_{n=1}^\infty A_n, \quad A_n := \{x \in A : f(x) > 1/n\}. \tag{46}$$

Since $\mu(A) > 0$, there is $n$ such that $\mu(A_n) > 0$. Otherwise, by subadditivity of measures,

$$\mu(A) \leq \sum_{n=1}^\infty \mu(A_n) = 0 \tag{47}$$

which contradicts the assumption $\mu(A) > 0$. It follows that

$$\int_A f \, d\mu \geq \int_{A_n} f \, d\mu \geq \int_{A_n} \frac{1}{n} \, d\mu = \frac{\mu(A_n)}{n} > 0. \tag{48}$$

This establishes the claim. $\square$

*Proof of Proposition 1.* Denote the measure of $(t, \boldsymbol{x}_t)$ where $t \sim \mathrm{unif}(0, 1)$ and $\boldsymbol{x}_t \sim \mathbb{Q}_t$ as $\mu$. (Assuming $\boldsymbol{D}_\theta$ can approximate a sufficiently large set of functions), define $\theta^*$ as the neural net parameter which satisfies

$$\boldsymbol{D}_{\theta^*}(\boldsymbol{x}_t, t) = \mathbb{E}_{\boldsymbol{x}_0 \sim \mathbb{Q}_{0|t}(\cdot | \boldsymbol{x}_t)}[\boldsymbol{x}_0] \tag{49}$$

for any $(\boldsymbol{x}_t, t)$ such that

$$\mathcal{L}_{\mathrm{GFM}}(\theta; \mathbb{Q}_{01}, \boldsymbol{x}_t, t) \geq \mathcal{L}_{\mathrm{GFM}}(\theta^*; \mathbb{Q}_{01}, \boldsymbol{x}_t, t) \tag{50}$$

or equivalently,

$$\mathcal{L}_{\mathrm{FM}}(\theta; \mathbb{Q}_{01}, \boldsymbol{x}_t, t) \geq \mathcal{L}_{\mathrm{FM}}(\theta^*; \mathbb{Q}_{01}, \boldsymbol{x}_t, t) \tag{51}$$

for any $(\boldsymbol{x}_t, t)$ and $\theta$ by Lemma 1.

We now show that a minimizer of Eq. (13) minimizes Eq. (1). Suppose $\theta$ minimizes Eq. (13), but there is a set $A$ with positive measure, *i.e.*, $\mu(A) > 0$, such that

$$\boldsymbol{D}_\theta(\boldsymbol{x}_t, t) \neq \mathbb{E}_{\boldsymbol{x}_0 \sim \mathbb{Q}_{0|t}(\cdot | \boldsymbol{x}_t)}[\boldsymbol{x}_0] \tag{52}$$

for all $(\boldsymbol{x}_t, t) \in A$. By Lemma 1,

$$\mathcal{L}_{\mathrm{GFM}}(\theta; \mathbb{Q}_{01}, \boldsymbol{x}_t, t) > \mathcal{L}_{\mathrm{GFM}}(\theta^*; \mathbb{Q}_{01}, \boldsymbol{x}_t, t) \tag{53}$$

for all $(\boldsymbol{x}_t, t) \in A$, and since $w(\boldsymbol{x}_t, t)$ and $d\mathbb{T}(t)$ are positive by assumption,

$$d\mathbb{T}(t) \cdot w(\boldsymbol{x}_t, t) \cdot \mathcal{L}_{\mathrm{GFM}}(\theta; \mathbb{Q}_{01}, \boldsymbol{x}_t, t) > d\mathbb{T}(t) \cdot w(\boldsymbol{x}_t, t) \cdot \mathcal{L}_{\mathrm{GFM}}(\theta^*; \mathbb{Q}_{01}, \boldsymbol{x}_t, t) \tag{54}$$

for all $(\boldsymbol{x}_t, t) \in A$, so by Lemma 2,

$$\mathbb{E}_{(t,\boldsymbol{x}_t) \sim \mu}[1_A(\boldsymbol{x}_t, t) \cdot d\mathbb{T}(t) \cdot w(\boldsymbol{x}_t, t) \cdot \mathcal{L}_{\mathrm{GFM}}(\theta; \mathbb{Q}_{01}, \boldsymbol{x}_t, t)] \tag{55}$$

$$> \mathbb{E}_{(t,\boldsymbol{x}_t) \sim \mu}[1_A(\boldsymbol{x}_t, t) \cdot d\mathbb{T}(t) \cdot w(\boldsymbol{x}_t, t) \cdot \mathcal{L}_{\mathrm{GFM}}(\theta^*; \mathbb{Q}_{01}, \boldsymbol{x}_t, t)] \tag{56}$$

where $1_A(\boldsymbol{x}_t, t) = 1$ if $(\boldsymbol{x}_t, t) \in A$ and $0$ if not, and so

$$\mathcal{L}_{\mathrm{GFM}}(\theta; \mathbb{Q}_{01}) = \mathbb{E}_{(t,\boldsymbol{x}_t) \sim \mu}[d\mathbb{T}(t) \cdot w(\boldsymbol{x}_t, t) \cdot \mathcal{L}_{\mathrm{GFM}}(\theta; \mathbb{Q}_{01}, \boldsymbol{x}_t, t)] \tag{57}$$

$$> \mathbb{E}_{(t,\boldsymbol{x}_t) \sim \mu}[d\mathbb{T}(t) \cdot w(\boldsymbol{x}_t, t) \cdot \mathcal{L}_{\mathrm{GFM}}(\theta^*; \mathbb{Q}_{01}, \boldsymbol{x}_t, t)] = \mathcal{L}_{\mathrm{GFM}}(\theta^*; \mathbb{Q}_{01}) \tag{58}$$

which contradicts the assumption that $\theta$ minimizes Eq. (13). It follows that if $\theta$ minimizes Eq. (13),

$$\boldsymbol{D}_\theta(\boldsymbol{x}_t, t) = \mathbb{E}_{\boldsymbol{x}_0 \sim \mathbb{Q}_{0|t}(\cdot|\boldsymbol{x}_t)}[\boldsymbol{x}_0] \tag{59}$$

almost everywhere w.r.t. $\mu$, it also minimizes

$$\mathcal{L}_{\mathrm{FM}}(\theta; \mathbb{Q}_{01}, \boldsymbol{x}_t, t) \tag{60}$$

almost everywhere w.r.t. $\mu$ by Lemma 1, which implies $\theta$ minimizes Eq. (1).

The other direction can be proven in an analogous manner. $\qquad\square$

## E.2 Proof of Proposition 2

*Proof of Proposition 2.* Let $\mathbb{Q}_{01}^0 = \mathbb{P}_0 \otimes \mathbb{P}_1$. If we assume zero initialization in output layer for $\boldsymbol{D}_\theta$,

$$\max_{\boldsymbol{x}_1} \mathcal{L}_{\mathrm{DM}}(\theta; \mathbb{Q}_{01}^0, \boldsymbol{x}_1, 1) / \min_{\boldsymbol{x}_1} \mathcal{L}_{\mathrm{DM}}(\theta; \mathbb{Q}_{01}^0 \boldsymbol{x}_1, 1) \tag{61}$$

$$= \max_{\boldsymbol{x}_1} \mathbb{E}_{\boldsymbol{x}_0 \sim \mathbb{Q}_{0|1}^0(\cdot|\boldsymbol{x}_1)} \|\boldsymbol{x}_0 - \boldsymbol{D}_\theta(\boldsymbol{x}_1, 1)\|_2^2 / \min_{\boldsymbol{x}_1} \mathbb{E}_{\boldsymbol{x}_0 \sim \mathbb{Q}_{0|1}^0(\cdot|\boldsymbol{x}_1)} \|\boldsymbol{x}_0 - \boldsymbol{D}_\theta(\boldsymbol{x}_1, 1)\|_2^2 \tag{62}$$

$$= \max_{\boldsymbol{x}_1} \mathbb{E}_{\boldsymbol{x}_0 \sim \mathbb{Q}_{0|1}^0(\cdot|\boldsymbol{x}_1)} \|\boldsymbol{x}_0\|_2^2 / \min_{\boldsymbol{x}_1} \mathbb{E}_{\boldsymbol{x}_0 \sim \mathbb{Q}_{0|1}^0(\cdot|\boldsymbol{x}_1)} \|\boldsymbol{x}_0\|_2^2 \tag{63}$$

$$= \max_{\boldsymbol{x}_1} \mathbb{E}_{\boldsymbol{x}_0 \sim \mathbb{P}^0} \|\boldsymbol{x}_0\|_2^2 / \min_{\boldsymbol{x}_1} \mathbb{E}_{\boldsymbol{x}_0 \sim \mathbb{P}^0} \|\boldsymbol{x}_0\|_2^2 = 1 \tag{64}$$

On the other hand, if we use a pre-trained diffusion model to initialize $\boldsymbol{D}_\theta$,

$$\boldsymbol{D}_\theta(\boldsymbol{x}_1, 1) = \boldsymbol{\mu}_0 \tag{65}$$

such that

$$\max_{\boldsymbol{x}_1} \mathcal{L}_{\mathrm{GFM}}(\theta; \mathbb{Q}_{01}^1, \boldsymbol{x}_1, 1) / \min_{\boldsymbol{x}_1} \mathcal{L}_{\mathrm{GFM}}(\theta; \mathbb{Q}_{01}^1, \boldsymbol{x}_1, 1) \tag{66}$$

$$= \max_{\boldsymbol{x}_1} \mathbb{E}_{\boldsymbol{x}_0 \sim \mathbb{Q}_{0|1}^1(\cdot|\boldsymbol{x}_1)} \|\boldsymbol{x}_0 - \boldsymbol{\mu}_0\|_2^2 / \min_{\boldsymbol{x}_1} \mathbb{E}_{\boldsymbol{x}_0 \sim \mathbb{Q}_{0|1}^1(\cdot|\boldsymbol{x}_1)} \|\boldsymbol{x}_0 - \boldsymbol{\mu}_0\|_2^2 \tag{67}$$

$$= \max_{\boldsymbol{x}_0} \|\boldsymbol{x}_0 - \boldsymbol{\mu}_0\|_2^2 / \min_{\boldsymbol{x}_0} \|\boldsymbol{x}_0 - \boldsymbol{\mu}_0\|_2^2 \tag{68}$$

because $\boldsymbol{x}_1 \mapsto \boldsymbol{x}_0 \sim \mathbb{Q}_{0|1}^1(\cdot|\boldsymbol{x}_1)$ is now a bijective map between $\mathbb{P}_0$ and $\mathbb{P}_1$ samples.

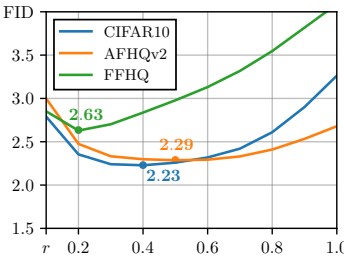 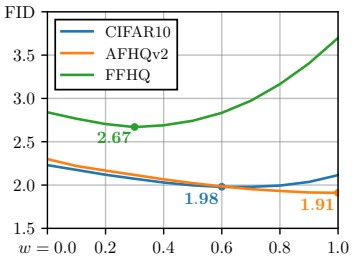

Figure 12: DPM-Solver $r$        Figure 13: AutoGuidance $w$

## F ADDITIONAL EXPERIMENTS

### F.1 LINEAR DISCRETIZATION LACKS DISCRIMINATIVE POWER

While all previous works use the uniform discretization to sample from ReFlow models, we use the sigmoid discretization to evaluate models in Sections 3.2 and 3.3. This is because, we found that the uniform discretization lacks discrimination power, *i.e.*, the ability to make the best of a given model, especially at small NFEs.

To demonstrate this, in Tab. 10, we re-evaluate models in Sec. 3.2.1 with the uniform discretization, and compare them with evaluation results with the sigmoid discretization with $\kappa = 10$. We observe that none of the FIDs with the uniform discretiza-

| $w(\boldsymbol{x}_t, t)$ | Uniform | Sigmoid |
|---|---|---|
| $1$ | **2.88** | 2.87 |
| $1/t$ | 2.89 | 2.76 |
| $1/t^2$ | 2.93 | **2.74** |
| $(\sigma^2 + 0.5^2)/(0.5\sigma)^2$ | 2.97 | 2.82 |
| $1/\mathbb{E}_{\boldsymbol{x}_t}[\mathrm{sg}[\mathcal{L}_{\mathrm{GFM}}(\theta; \mathbb{Q}_{01}, \boldsymbol{x}_t, t)]]$ | 2.98 | 2.79 |
| $1/\mathrm{sg}[\mathcal{L}_{\mathrm{GFM}}(\theta; \mathbb{Q}_{01}, \boldsymbol{x}_t, t)]$ | 2.95 | **2.74** |

Table 10: Uniform vs. sigmoid ($\kappa = 10$) discretizations with Heun on AFHQv2.

tion are better than the worst FID with the sigmoid discretization. Moreover, the model with our proposed weight, when evaluated with the uniform schedule, performs worse than the model with uniform weight.

We speculate this happens because, as analyzed in Sec. 3.4, large curvature regions for ReFlow ODEs occur near $t \in \{0, 1\}$, but the uniform discretization fails to account for them. So, the uniform discretization is unable to accurately capture the differences in ODE trajectories between different models. Due to these reasons, we opt to use the sigmoid discretization to distinguish training techniques that work from those that do not.

### F.2 DPM-SOLVER AND GUIDANCE ABLATIONS

Recall that the DPM-Solver update for Eq. (3) is given as

$$\boldsymbol{x}_{t_i} \leftarrow \boldsymbol{x}_{t_{i+1}} + (t_i - t_{i+1})(\tfrac{1}{2r}\boldsymbol{v}_\theta(\boldsymbol{x}_{s_{i+1}}, s_{i+1}) + (1 - \tfrac{1}{2r})\boldsymbol{v}_\theta(\boldsymbol{x}_{t_{i+1}}, t_{i+1})). \quad (69)$$

In Fig. 12, we show the FID for various values of $r \in (0, 1]$. While we can get better FIDs than those in Tab. 7 by using $r$ tailored to individual datasets, we opt for simplicity and set $r = 0.4$ as our improved choice, which still yields better FID than the Heun solver, *i.e.*, using $r = 1$.

For conditional ReFlow models, classifier-free guidance (CFG) (Ho & Salimans, 2022) can be formulated as solving the ODE

$$d\boldsymbol{x}_t = \{(1 + w) \cdot \boldsymbol{v}_\theta(\boldsymbol{x}_t, t, c) - w \cdot \boldsymbol{v}_\theta(\boldsymbol{x}_t, t, \varnothing)\}\, dt \quad (70)$$

where $\boldsymbol{v}_\theta(\boldsymbol{x}_t, t, c)$ is velocity conditioned on $c$, and $\boldsymbol{v}_\theta(\boldsymbol{x}_t, t, \varnothing)$ is an unconditional velocity, and $w$ is guidance scale. Note that $w = 0$ reduces the ODE to standard class-conditional generation. In practice, we train conditional velocities with label dropout such that $\boldsymbol{v}_\theta(\boldsymbol{x}_t, t, c)$ and $\boldsymbol{v}_\theta(\boldsymbol{x}_t, t, \varnothing)$ can be evaluated in parallel, by passing class labels to the former and null labels to the latter.

For unconditional ReFlow models, AutoGuidance (Karras et al., 2024) can be formulated as solving

$$d\boldsymbol{x}_t = \{(1 + w) \cdot \boldsymbol{v}_\theta(\boldsymbol{x}_t, t) - w \cdot \hat{\boldsymbol{v}}_\phi(\boldsymbol{x}_t, t)\}\, dt \quad (71)$$

where $\hat{\boldsymbol{v}}_\phi$ is a degraded version of $\boldsymbol{v}_\theta$. In practice, we use ReFlow models trained with the baseline training configuration (see Tab. 1) for $10k$ iterations as $\hat{\boldsymbol{v}}_\phi$. While other choices of $\hat{\boldsymbol{v}}_\phi$ may offer better FIDs, as AG is not the main topic of our paper, we do not perform an extensive search.

### F.3 QUANTIFYING REFLOW BIAS REDUCTION

The extent of the bias introduced by ReFlow, and its reduction with our techniques can be calculated by comparing Tables 6, 7, 8. Specifically, since we measure the performance of our ReFlow models via Frechet Inception Distance (Wasserstein-2 distance between Gaussian approximations of true and model distributions in the feature space of the Inception network) (Heusel et al., 2017), we may use

$$\text{(FID of } \mathbb{Q}_0^{n+1}) - \text{(FID of } \mathbb{Q}_0^{n})$$

|  | CIFAR10 | AFHQv2 | FFHQ |
|---|---|---|---|
| BSL | 0.86 | 0.91 | 1.89 |
| DYN | 0.61 | 0.59 | 1.30 |
| DYN+LRN | 0.41 | 0.51 | 0.74 |
| DYN+LRN+INF | 0.26 | 0.34 | 0.45 |
| DYN+LRN+INF+GD | 0.01 | −0.05 | 0.28 |

Table 11: Amount of bias introduced by Re-Flow under different settings. Negative bias means our model achieves a better FID than the diffusion model used to generate $\mathbb{Q}_{01}^1$.

as a proxy for the amount of bias introduced by Re-Flow. In our work, with $n = 1$, $\mathbb{Q}_0^1$ is the EDM model marginal and $\mathbb{Q}_0^2$ is our ReFlow model marginal. In Table 11, we summarize the extent of bias as we add improved training dynamics (DYN), improved learning (LRN), improved inference (INF), and guidance (GD) to the baseline (BSL) setting. With everything combined, our techniques achieve a significant reduction in bias.

## F.4 SAMPLE VISUALIZATION

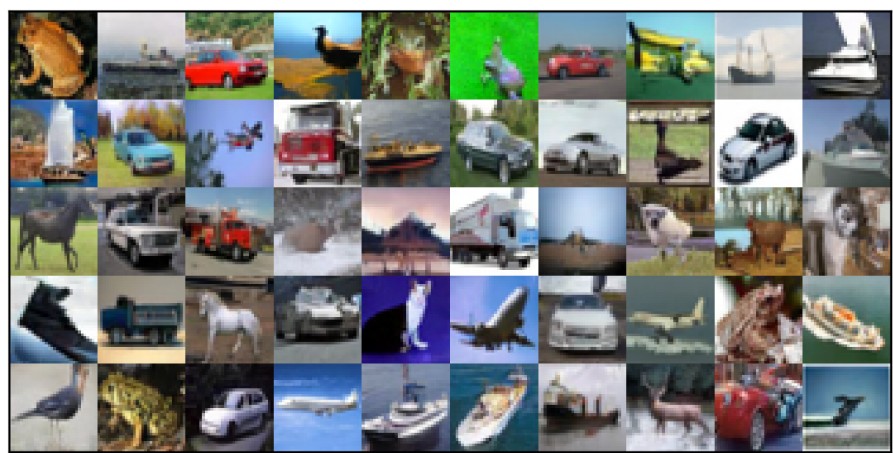

(a) BSL, 2.83 FID with 9 NFEs

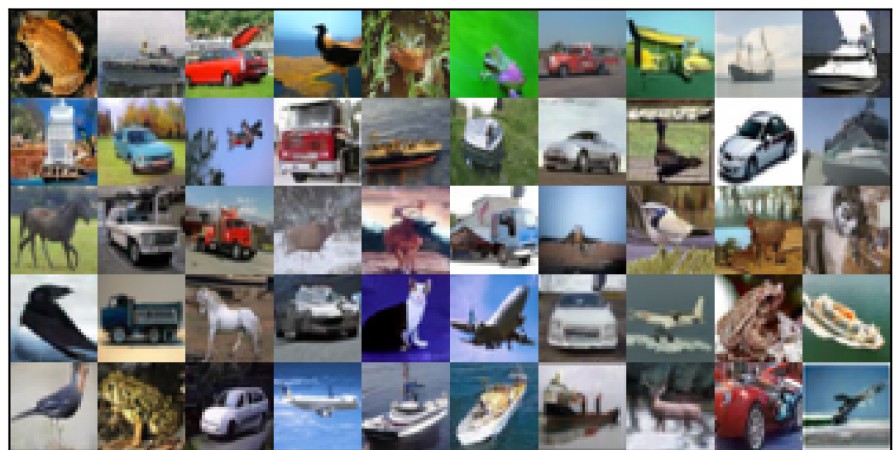

(b) DYN+LRN+INF, 2.23 FID with 9 NFEs

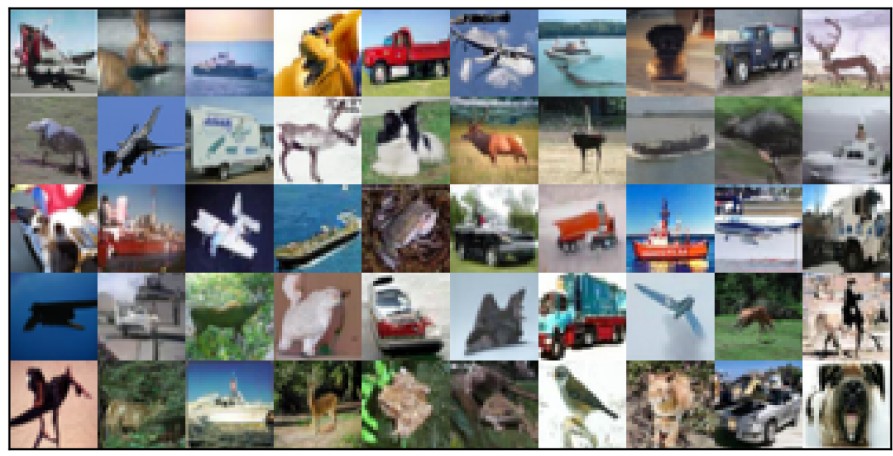

(c) DYN+LRN+INF+AG, 1.98 FID with 9 NFEs

Figure 14: CIFAR10 samples with fixed random seeds

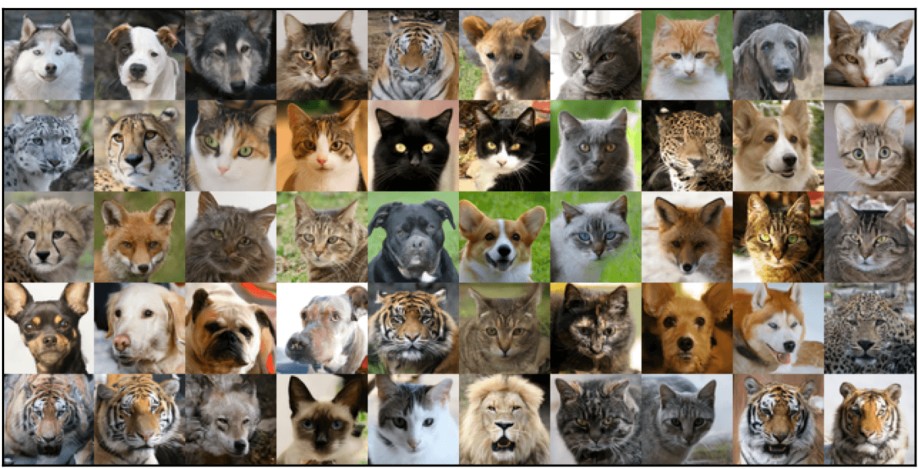

(a) BSL, 2.87 FID with 9 NFEs

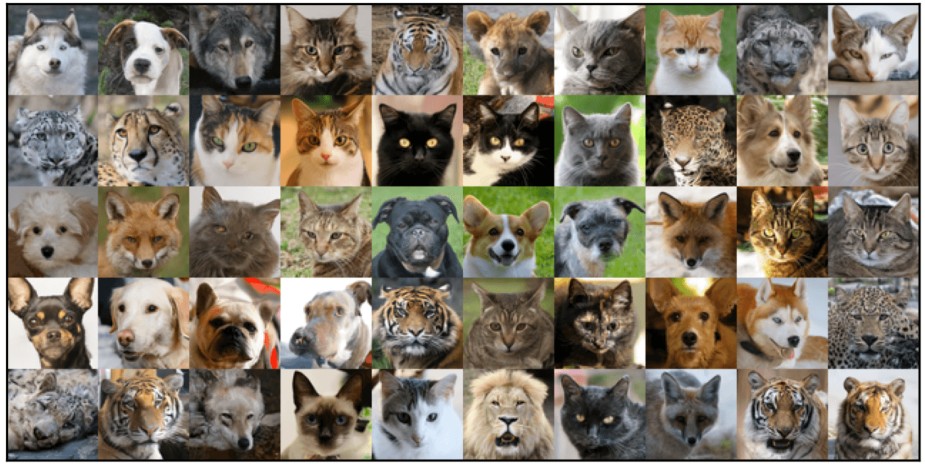

(b) DYN+LRN+INF, 2.30 FID with 9 NFEs

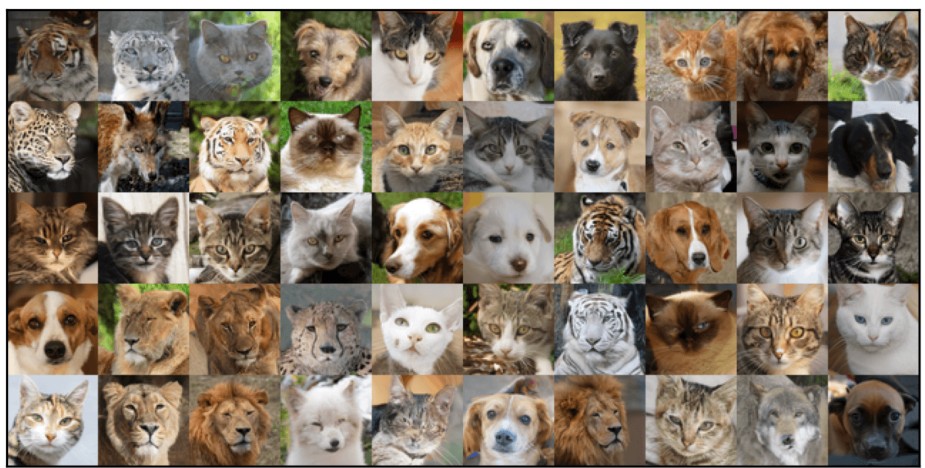

(c) DYN+LRN+INF+AG, 1.91 FID with 9 NFEs

Figure 15: AFHQv2 samples with fixed random seeds

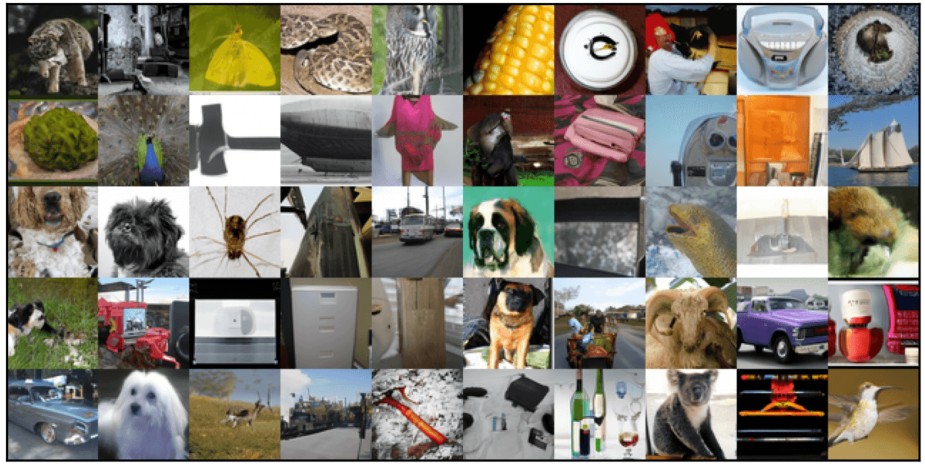

(a) BSL, 4.27 FID with 9 NFEs

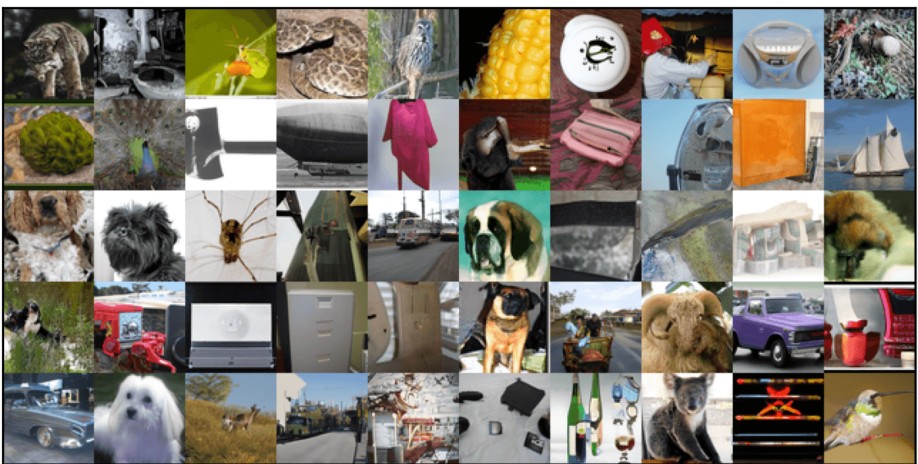

(b) DYN+LRN+INF, 3.49 FID with 9 NFEs

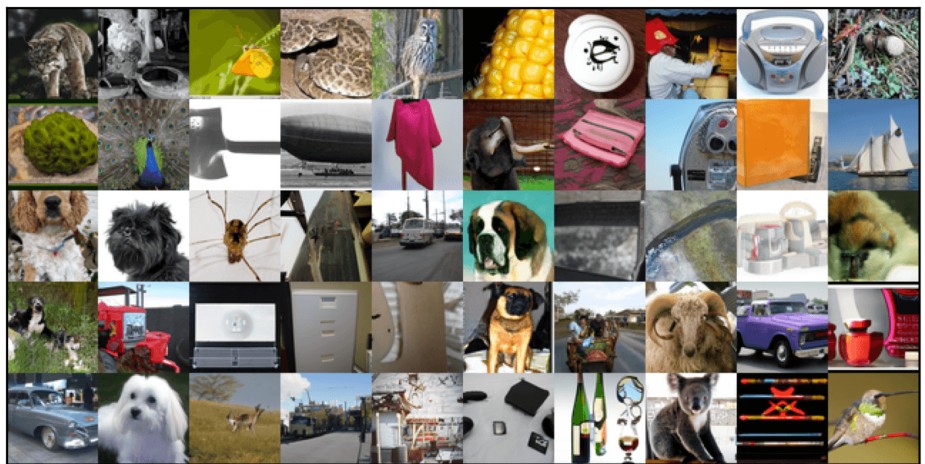

(c) DYN+LRN+INF+CFG, 1.74 FID with 9 NFEs

Figure 16: ImageNet-64 samples with fixed random seeds

## G  FURTHER DISCUSSION OF OUR IMPROVEMENTS

Here, we provide further insight into why our techniques improve upon prior practice.

**Loss normalization.**  In Section 3.1 we show that the Generalized Flow Matching objective is a collection of regression problems aggregated over $(\boldsymbol{x}_t, t)$, and at Section 3.2.1, we theoretically and numerically show that individual regression loss values during ReFlow can have vastly different scales with respect to $(\boldsymbol{x}_t, t)$. Multi-task learning interpretation of loss normalization (Zhang et al., 2018; Karras et al., 2023b) along with our observation motivates loss normalization with respect to both $\boldsymbol{x}_t$ and $t$, and in Table 2, we demonstrate that loss normalization beats all other weights.

**Time distribution.**  In Section 3.2.2, we explain that previous work (Lee et al., 2024) uses the cosh time distribution in order to oversample $t$ near 0 or 1, where most of the learning happens. We also explain that we choose the increasing exponential time distribution, since our weight function already compensates for vanishing loss near $t = 0$.

**Loss function.**  In Section 3.2.3, we show that using $\phi$ in the loss function is equivalent to preconditioning the loss gradient, and it is well known that an appropriate gradient preconditioning can accelerate model convergence (Kingma & Ba, 2015).

**Dropout.**  In Section 3.3.1, we explain that we need models with larger Lipschitz constants if we wish to learn better ReFlow models. This is because ReFlow converges to a straight ODE, and a straight ODE is ultimately a push-forward generative model, and (Salmona et al., 2022) formally shows that a push-forward generative model needs to have a Lipschitz constant in order to map a unimodal distribution to a multi-modal distribution accurately. For instance, Corollaries 5, 6, 8 in (Salmona et al., 2022) show divergence or distance between data and model distributions is lower bounded by a decreasing function of Lipschitz constant of the push-forward model. This motivates us to increase effective model capacity by decreasing dropout probability.

**Training data.**  Alemohammad et al. (2024) shows that recursively training generative models on data generated by itself reduces the quality and diversity of data. Alemohammad et al. (2024) also shows one can delay or prevent degradation by injecting real data into the training loop. Using forward pairs can be interpreted as an instance of injecting real data in the training loop, and using projected pairs can be interpreted as synthesizing new real data by solving the projection problem.

**Discretization.**  In Figure 8, as evidenced by truncation error for the uniform discretization, the ODE after ReFlow has high curvature regions near $t = 0$ and 1. Truncation error for our sigmoid discretization schedule shows it is able to effectively control the error at the extremes of the interval.

**Solver.**  We note that Karras et al. (2022) popularized Heun as an alternative to Euler, among the large set of solvers considered, based primarily on strong empirical performance. We argue that DPM-solver is a generalization of Heun (coinciding for $r = 1$), and we observe that setting $r = 0.4$ performs better than Heun's second order solver. $r$ can be tuned cheaply, especially since our ReFlow models produce state-of-the-art results with NFE $< 10$.

