# OpenReview forum: "Simple ReFlow: Improved Techniques for Fast Flow Models"
_ICLR.cc/2025/Conference — ICLR 2025 Poster_

### Official Review · Reviewer_VmAn · 2024-10-21

**Soundness:** 3
**Presentation:** 3
**Contribution:** 3
**Rating:** 8
**Confidence:** 5

**Summary:**

This work addresses the performance degradation of ReFlow without violating its theoretical foundations. ReFlow aims to straighten the trajectories of ordinary differential equations (ODEs) by flow-matching between marginal distributions connected by a previously trained flow ODE. Theoretically, an infinite number of ReFlow updates should result in a straight trajectory that enables perfect translation between marginals with a single function evaluation. However, in practice, ReFlow suffers from a drop in sample quality, and previous attempts to mitigate this have relied on heuristic methods like perceptual losses, which deviate from the theoretical framework and may not preserve the underlying probability distributions.

The authors generalize the ReFlow training loss and categorize design choices into three key groups: training dynamics, learning, and inference. They analyze previous practices, identify potential pitfalls, and propose improved techniques within the theoretical bounds. Extensive ablation studies on datasets such as CIFAR10, FFHQ, and AFHQv2 demonstrate that their methods consistently improve sample quality and Fréchet Inception Distance (FID) scores. Ultimately, they achieve state-of-the-art results in fast neural ODE-based generation without relying on heuristic methods, offering a theoretically sound and practical approach to accelerating generative models.

**Strengths:**

- The paper does a very good job at ablating design choices to improve ReFlow.
- The experimental results are state-of-the-art, by a good margin.
- The paper can be very helpful for practitioners that face similar choices when training their models.

**Weaknesses:**

- Each of their contributions may be labeled as incremental, which means that the whole paper could also be labeled as incremental. Regardless, it is a very valuable paper to the community because it sheds light on several empirical choices one must make.

- The statement “This again ensures that the loss is minimized when $D_{\theta}$
outputs the posterior mean” in line 184 is false. In fact, what holds is
$D^*(x_t,t) = \phi^{-1}(\mathbb{E}_{x_0 \sim Q(\cdot|x_t)} [\phi(x_0)])$.
When $\phi$ is far from the identity, $D^*(x_t,t)$ may be far from the desired point. This does not makes the empirical findings less valuable, but it does invalidate the theoretical claims that the authors make about the training loss based on $\phi$ adhering to FM theory.

**Questions:**

- As described in lines 100-111, the original ReFlow algorithm works by iteratively training $D^{n+1}$ using the coupling induced by $D^{n}$. Does Simple ReFlow proceed analogously? If so, how many iterations are used? It would be good to include this information in the paper.

- The authors propose two improvements with regards to training data: forward pairs and projected pairs. Hence, there are three ways to obtain trajectories: backward pairs, forward pairs and projected pairs. How are the three approaches combined? For example, what is the approach used to obtain the results in Table 8?

- The projected pairs approach is very interesting. The authors propose one way to solve the projection problem, which involves recasting the constraint on the marginals of $\Gamma_{01}$ as a penalty term, which forces them to use GANs. It seems that solving a Schrodinger bridge problem between P_0 and P_1 with an appropriate regularization term would also achieve the final goal of the paper, but of course one then loses the connection with ReFlow. Are there other alternatives that may be simpler? Also, it would be good to provide a more detailed explanation in Subsec. D.3, e.g., more details in how problem 42 is solved.

- What do the authors think is the improvement that is most responsible for the performance boost that is observed? As a reviewer, I think the training data improvements may be the most critical ones, but it would be nice to comment on that in the Conclusion section.

---

> ### Author Response · Authors · 2024-11-20
> **Reply to Reviewer VmAn**
>
> We sincerely thank Reviewer VmAn for the insightful review. Here, we answer the Reviewer's concerns and questions.
>
> > **... a very valuable paper to the community because it sheds light on several empirical choices one must make.**
>
> We appreciate that the Reviewer recognizes the value of our contributions.
>
> > **[W1] The statement “This again ensures that the loss is minimized when $D_\theta$ outputs the posterior mean” in line 184 is false. In fact, what holds is $D(x_t,t) = \phi^{-1}(\mathbb{E}[\phi(x_0)])$. When phi is far from the identity, $D(x_t,t)$ may be far from the desired point. This does not make the empirical findings less valuable, but it does invalidate the theoretical claims that the authors make about the training loss based on phi adhering to FM theory.**
>
> We restrict $\phi$ to be an invertible linear map. As $\phi$ is linear, we may exchange $\mathbb{E}$ and $\phi$, and given $\phi$ is invertible, $\phi^{-1}$ exists, and cancels $\phi$, yielding $D(x_t,t) = \mathbb{E}[x_0]$. Specifically,
>
> $$D(x_t,t) = \phi^{-1}(\mathbb{E}[\phi(x_0)]) = \phi^{-1}(\phi(\mathbb{E}[x_0])) = \mathbb{E}[x_0].$$
>
> > **[Q1] As described in lines 100-111, the original ReFlow algorithm works by iteratively training $D^{n+1}$ using the coupling induced by $D^n$. Does Simple ReFlow proceed analogously? If so, how many iterations are used? It would be good to include this information in the paper.**
>
> We use one iteration of ReFlow, i.e., report results for $D^2$ given $D^1$. We have clarified this point further in our revised paper (**line 214**)
>
> > **[Q2] The authors propose two improvements with regards to training data: forward pairs and projected pairs. Hence, there are three ways to obtain trajectories: backward pairs, forward pairs and projected pairs. How are the three approaches combined? For example, what is the approach used to obtain the results in Table 8?**
>
> Details for forward pairs are described in Section 3.3.2. When we use projected pairs, we do not mix them with any forward or backward pairs. We have clarified this point in our revised paper (**line 426**). We have also described hyper-parameters for Table 8 results in **Appendix D.1.1**.
>
> > **[Q3] The projected pairs approach is very interesting. The authors propose one way to solve the projection problem, which involves recasting the constraint on the marginals of Gamma01 as a penalty term, which forces them to use GANs. … Are there other alternatives that may be simpler? Also, it would be good to provide a more detailed explanation in Subsec. D.3, e.g., more details on how problem 42 is solved.**
>
> A cheaper alternative would be to use forward pairs, which we have observed to provide similar performance improvements (see Tab. 6). We have also added pseudo-codes for solving the projection problem in the revised version of our paper (**Alg. 1 and 2 in Appendix D.3**).
>
> > **It seems that solving a Schrodinger bridge problem between P_0 and P_1 with an appropriate regularization term would also achieve the final goal of the paper, but of course one then loses the connection with ReFlow.**
>
> ReFlow can be seen as a particular case of the Iterative Markovian Fitting (IMF) procedure which is used to obtain a Schrödinger Bridge (SB) [1]. IMF generally uses a Brownian bridge for interpolating between marginals during training, and setting the standard deviation in the Brownian bridge to 0 recovers a linear interpolation and hence the ReFlow procedure.
>
> The advantage of ReFlow over SB, is in obtaining straight trajectories which lead to faster sampling, and no need to train two networks as in IMF. Simulating a SB is typically stochastic, one can obtain a probability flow ODE from SB which may perhaps be smoother but less likely to be as straight as from the ReFlow procedure. Unfortunately this means that learning a SB is perhaps just as difficult as the ReFlow procedure, and may also have some simulation error.
>
> We tried projecting the marginals via Sinkhorn (equivalent to static SB for discrete marginals) and have shown these results in the paper.
>
> [1] Shi et al., Diffusion Schrödinger Bridge Matching, 2023.
>
> > **[Q3] What do the authors think is the improvement that is most responsible for the performance boost that is observed? As a reviewer, I think the training data improvements may be the most critical ones, but it would be nice to comment on that in the Conclusion section.**
>
> Each set of contributions can be related as improving training of the initial flow will improve the quality of the samples within the coupling used as training data. This has a similar outcome to directly improving the training data via projections or forward pairs.
>
> We have added the following sentence to the conclusion: _In terms of FID score, the weight function and smaller dropout probability contributed the most to the performance boost. However, in terms of novelty, we believe training data improvements and the generalized loss function are our largest contributions._

---

> > ### Comment · Reviewer_VmAn · 2024-11-26
> > **Reply to the authors' response**
> >
> > Thank you for the effort. I keep my score.

---

### Official Review · Reviewer_Zbb5 · 2024-11-01

**Soundness:** 4
**Presentation:** 3
**Contribution:** 3
**Rating:** 6
**Confidence:** 4

**Summary:**

This paper proposes several improvements to training dynamics, learning, and inference for Rectified Flow models (e.g. initialization from the EDM model, loss reweighting, new loss function, refinements on noise-image pairs, etc.). Thorough ablation studies verify the effectiveness of the proposed improvements and the final experiment results are competitive among diffusion and flow-based models for few-steps sampling.

**Strengths:**

1. The paper is well-written and easy to follow, with each step of the ablation study clearly described and the results effectively presented.
2. The final experiment results after combining all improvements show lower FID and lower straightness compared with other diffusion-based and flow-based methods.

**Weaknesses:**

1. The projected pairs operation, which projects the coupling $Q_{01}^1$ to $\Pi(P_0, P_1)$, complicates the training pipeline (GAN training and solving Sinkhorn) while yielding only marginal performance improvements, as indicated in Table 6 (with a maximum FID improvement of 0.02, which is not even statistically significant).

2. The authors frequently use the notation 'OT ODE' and 'OT map' throughout the paper. However, as noted in references [1] and [2], the iterative rectified flow does not generally converge to the optimal transport mapping in dimensions higher than or equal to 2. Furthermore, the algorithm presented in this paper corresponds to the 2-rectified flow. Therefore, the use of the 'OT' notation is not appropriate.

[1] Rectified Flow: A Marginal Preserving Approach to Optimal Transport. arxiv 2209.14577

[2] Flow Straight and Fast: Learning to Generate and Transfer Data with Rectified Flow. arxiv 2209.03003

**Questions:**

I do not have other questions.

---

> ### Author Response · Authors · 2024-11-20
> **Reply to Reviewer Zbb5**
>
> We deeply appreciate Reviewer Zbb5's constructive comments. Here, we address the Reviewer's concerns.
>
> > **[W1] The projected pairs operation, which projects the coupling $\mathbb{Q}_{01}^1$  to $\Pi(\mathbb{P}_0,\mathbb{P}_1)$, complicates the training pipeline (GAN training and solving Sinkhorn) while yielding only marginal performance improvements, as indicated in Table 6 (with a maximum FID improvement of 0.02, which is not even statistically significant).**
>
> As noted in Section 3.3.2, it may be necessary to use improved dynamics in order to get the best out of improved coupling. Without improved dynamics, projected pairs offer FID improvements of 0.06, 0.05, 0.02 but when combined with improved dynamics, we get larger improvements of 0.05, 0.06, 0.04.
>
> We also note that we already provide a cheaper alternative to projected pairs, i.e., forward pairs, which offer similar performance improvements but without GAN and solving Sinkhorn. We have verified the scalability of forward pairs on ImageNet-64.
>
> > **[W2] The authors frequently use the notation 'OT ODE' and 'OT map' throughout the paper. However, as noted in references [1] and [2], the iterative rectified flow does not generally converge to the optimal transport mapping in dimensions higher than or equal to 2. Furthermore, the algorithm presented in this paper corresponds to the 2-rectified flow. Therefore, the use of the 'OT' notation is not appropriate.**
>
> We thank the Reviewer for pointing this out. We agree with the reviewer that this was inaccurate, and replaced any references to “OT ODE” throughout the main text by “straight ODE” (**lines 111-112, 288, 366, 368-369, 449**). Indeed, as detailed in [1], further conditions are required for ReFlow to converge to OT (conservative vector fields). The emphasis of our contributions is not in relation to OT, and have moved and enhanced this discussion in the Appendix A.
>
> [1] Rectified Flow: A Marginal Preserving Approach to Optimal Transport. arxiv 2209.14577
>
> [2] Flow Straight and Fast: Learning to Generate and Transfer Data with Rectified Flow. arxiv 2209.03003

---

> > ### Comment · Reviewer_Zbb5 · 2024-11-21
> >
> > I am grateful to the authors for taking the time to provide the response. I appreciate that the authors have realized the inappropriate use of the optimal transport notation. I will maintain my original rating.

---

> > > ### Comment · Reviewer_7gZz · 2024-11-21
> > >
> > > > I appreciate that the authors have realized the inappropriate use of the optimal transport notation. I will maintain my original rating.
> > >
> > > Just quickly commenting in here that this was simply an oversight in the writing, and it was clear the paper/authors meant the straight ODE. This was completely resolved in the new version of the paper, and does not affect the correctness of any of the derivations, methodology, or experimental results

---

> ### Author Response · Authors · 2024-11-25
> **Have we addressed your concerns and questions?**
>
> Many thanks for having taken the time to read our rebuttal, and more generally for providing very useful comments that have helped us improve the presentation of our paper, by resolving any ambiguities regarding OT ODE vs. straight ODE.
>
> Please let us know if there are any outstanding concerns we can address.

---

### Official Review · Reviewer_3r18 · 2024-11-03

**Soundness:** 3
**Presentation:** 3
**Contribution:** 2
**Rating:** 6
**Confidence:** 4

**Summary:**

This paper provides an empirical study of how to improve the FID of rectified flow based image generative models. They aggregate a number of design choices studied in the context of diffusion and reconsider them in the context of flow matching and rectification. These design choices focus on the various aspects of the loss function: the metric under which the regression is performed (e.g. MSE vs PIPs, vs generalized invertible mappings), the time weighting, and loss normalization. They then consider training choices, like how the training data is chosen based on the first deterministic coupling of the rectification (fixing a Gaussian base value for each data point, called forward pairs). Finally they run ablations over discretizations of the ODE.

**Strengths:**

The paper provides a number of useful empirical studies of what ultimately influences the performance of the flow matching models and when they are rectified to straighter paths. As the reviewer sees it, the paper aggregates many of the design space choices that arise from the diffusion model literature and uses them to study how these knobs affect flow matching. It is nice to have them all considered in one place.

The inclusion of a new high pass filter metric on the loss is nice, and could be well motivated in other domains where we explicitly know these frequency regimes are important, e.g. for generative models in the sciences.

The extent of the ablations is thorough and the reviewer acknowledges the size of this experimental undertaking!

**Weaknesses:**

Thanks to the authors for their hard work on this project.

*Main remarks*:
- While the paper provides a number of empirical insights for training, it would be nice if there was more of an analysis of why these tricks ultimately help. The reviewer stresses this a bit because a number of the proposed innovations are directly inspired from their application in other work, e.g. the improvement of loss normalization in Karras et al (2023b), and e.g. equations (8)-(12).

- The reviewer is confused by the reference to OT solutions throughout, e.g. in Section 3.3.1, which suggests a potential misconception about the rectified flow procedure. In particular, as the reviewer understands it, one iteration does not give OT, but instead gives a monge map, in the sense that the coupling $P(x_0, x_1)  = P_0(x_0) \delta(x - T(x_0))$, where $T$ is the transport endowed from learning the velocity field first with $x_0, x_1$ independently drawn in the loss. This is certainly not OT (brenier polar factorization theorem), but is a deterministic coupling. Can the authors address this?

- The argument of proposition 2 is confusing, perhaps due to a typo. Is the first line supposed to specify the DM loss and not the GFM loss? Currently they are both statements about GFM.

- These analyses are mostly limited to the image domain, and it's hard to understand how any of these

- Can the authors address how large the bias is in rectification under the different procedures somehow? By this, the reviewer means that each iterate of rectification introduces a bias unless the flow was initially learned perfectly. The reviewer understands this to be difficult to measure, but it would be nice to understand how these ablations affect or relate to the extent of this bias. This might sit better in the questions section.




*Small remarks*:
-  the fundamental idea of of FM and rectified flow are equivalent to the method of stochastic interpolants, and should probably be cited accordingly.
- There is in a sense a competing small set of remarks about the benefits of rectification vs distillation in the appendix section C, saying that there are benefits of rectification over distillation because rectification maintains an ODE-like structure. But this ignores the fact that the learning problem after a rectification is a distillation problem...and it too is now fitting to an imperfect teacher model. I'm not sure this remark really contributes much to the discussion.

*Very small remarks*: A few typos to watch out for.
- "we  we" top of page 5

**Questions:**

The reviewer is trying to understand the language of the following claim in 3.3.1:

"Our improvement. We observe that learning the OT ODE is a harder task than learning the diffusion
probability flow ODE. For instance, at t = 1, the optimal DM denoiser only needs to predict the data mean $\mathbb E_{x_0 \sim P_0} [ x_0] $ for any input $x_1 ∼ P_1$, but an optimal ReFlow denoiser has to directly map noise to image along the bijective OT map."

Should the reviewer interpret this as being the claim that: The velocity field solving the probability flow is generically for $\mathbb E[\dot x_t | x_t = x]$, and for a deterministic coupling $P(x_0, x_1) = P_0 (x_0) \delta (x - T(x_0)$, this conditional expectation collapses because only one $x_t$ passes through $x$? This would be true for any monge coupling, and not just the OT coupling.

---

> ### Author Response · Authors · 2024-11-20
> **Reply to Reviewer 3r18 (Part 1/4)**
>
> We are grateful for Reviewer 3r18's detailed and constructive feedback. We address the Reviewer's concerns and questions below.
>
> > **[W1] While the paper provides a number of empirical insights for training, it would be nice if there was more of an analysis of why these tricks ultimately help. The reviewer stresses this a bit because a number of the proposed innovations are directly inspired from their application in other work, e.g. the improvement of loss normalization in Karras et al (2023b), and e.g. equations (8)-(12).**
>
> We thank the reviewer for their comment, given this is central to our contributions we want to ensure our reasoning is understood. Our improvements are grounded upon analyses or intuitions either presented in our work or in a previous work. We clarify this point below with further discussion of each of our improvements, which is now included in the main text and  **Appendix G** of the revised version of our paper.
>
> - **Loss normalization.** In Section 3 we show that the Generalized Flow Matching objective is a collection of regression problems aggregated over $(x_t,t)$, and at Section 3.2.1, we theoretically and numerically show that individual regression loss values during ReFlow can have vastly different scales with respect to $(x_t,t)$. Multi-task learning interpretation of loss normalization [1,2] along with our observation motivates loss normalization with respect to both $x_t$ and $t$, and in Table 2, we demonstrate that loss normalization beats all other weights.
>
> - **Time distribution.** In Section 3.2.2, we explain that previous work [3] uses the cosh time distribution in order to oversample $t$ near 0 or 1, where most of the learning happens. We also explain that we choose the increasing exponential time distribution, since our weight function already compensates for vanishing loss near $t = 0$.
>
> - **Loss function.** In Section 3.2.3, we show that using $\phi$ in the loss function is equivalent to preconditioning the loss gradient, and it is well known that an appropriate gradient preconditioning can accelerate model convergence [4].
>
> - **Dropout.** In Section 3.3.1, we explain that we need models with larger Lipschitz constants if we wish to learn better ReFlow models. This is because ReFlow converges to a straight ODE, and a straight ODE is ultimately a push-forward generative model, and [5] formally shows that a push-forward generative model needs to have a Lipschitz constant in order to map a unimodal distribution to a multi-modal distribution accurately (for instance, Corollaries 5, 6, 8 in [5] show divergence or distance between data and model distributions is lower bounded by a decreasing function of Lipschitz constant of the push-forward model). This motivates us to increase effective model capacity by decreasing dropout probability.
>
> - **Training data.** [6] shows that recursively training generative models on data generated by itself reduces the quality and diversity of data. [6] also shows one can delay or prevent degradation by injecting real data into the training loop. Using forward pairs can be interpreted as an instance of injecting real data in the training loop, and using projected pairs can be interpreted as synthesizing new real data by solving the projection problem.
>
> - **Discretization.** In Figure 8, as evidenced by truncation error for the uniform discretization, the ODE after ReFlow has high curvature regions near $t = 0$ and 1. Truncation error for our sigmoid discretization schedule shows it is able to effectively control the error at the extremes of the time interval.
>
> - **Solver.** We note that [7] popularized Heun as an alternative to Euler, among the large set of solvers considered, based primarily on strong empirical performance. We argue that DPM-solver is a generalization of Heun (coinciding for $r=1$), and we observe that setting $r = 0.4$ performs better than Heun’s second order solver. $r$ can be tuned cheaply, especially since our ReFlow models produce state-of-the-art results with NFE < 10
>
> [1] Kendall et al., “Multi-task learning using uncertainty to weigh losses for scene geometry and semantics”, CVPR, 2018.
>
> [2] Karras et al., “Analyzing and Improving the Training Dynamics of Diffusion Models”, CVPR, 2024.
>
> [3] Lee et al., “Improving the Training of Rectified Flows”, NeurIPS, 2024.
>
> [4] Kingma et al., “Adam: A Method for Stochastic Optimization”, ICLR, 2015.
>
> [5] Salmona et al., “Can Push-forward Generative Models Fit Multimodal Distributions?”, NeurIPS, 2022.
>
> [6] Alemohammad et al., “Self-Consuming Generative Models Go MAD”, ICLR, 2024.
>
> [7] Karras et al., “Elucidating the Design Space of Diffusion-Based Generative Models”, NeurIPS, 2022.

---

> > ### Comment · Reviewer_3r18 · 2024-12-01
> >
> > Thanks for the thorough replies. I have bumped my score accordingly!

---

> > > ### Author Response · Authors · 2024-12-02
> > >
> > > Thank you for increasing the score! We remain available for further questions until the end of the extended discussion period.

---

> ### Author Response · Authors · 2024-11-20
> **Reply to Reviewer 3r18 (Part 2/4)**
>
> > **[W2] .. confused by the reference to OT solutions throughout ... one iteration does not give OT, but instead gives a monge map, in the sense that the coupling $P(x_0,x_1)=P_0(x_0)\delta(x-T(x_0))$, where $T$ is the transport endowed from learning the velocity field first with $x_0,x_1$  independently drawn in the loss. This is certainly not OT (brenier polar factorization theorem), but is a deterministic coupling. Can the authors address this?**
>
> We thank the Reviewer for pointing out this ambiguity. This was an abuse of terminology and has been corrected. In our revised paper, we have removed direct references to OT, and replaced ‘‘OT ODE" with "straight ODE" throughout (**lines 111-112, 288, 366, 368-369, 449**).
>
> While we agree that this wording was confusing, we would like to emphasize that the objective of our paper (and more generally Reflow procedures) is **not to obtain an OT map**, but use methods related to OT for faster sampling.
>
> In the Section 5.4 of the seminal work “A Marginal Preserving Approach to Optimal Transport” [1] you will find a  detailed exposition of how ReFlow relates to OT. We pushed this discussion in the appendix of the original submission. We have further added to this section and hope that this clarifies the confusion.
>
> As you point out, our procedure (and Reflow procedures in general) do result in a deterministic map, but are not guaranteed to be the optimal transport map. Similar to OT, ReFlow targets a straight path. If learnt correctly the ReFlow procedure will reduce the transport cost for any convex ground cost at each reflow iteration. This is a stronger outcome than OT with a fixed ground cost. However, as per [1] Sec. 5.4, one may enforce convergence to OT and hence Monge map for squared Euclidean ground cost via ReFlow by setting the learnt velocity vector field to be conservative. We are not aware of quantitative convergence rates of ReFlow to OT; but indeed the number of ReFlow iterations may be more than 1 or 2, depending on the problem at hand.
>
> [1] Liu, A Marginal Preserving Approach to Optimal Transport, arXiv, 2022.
>
> > **[W3] The argument of proposition 2 is confusing, perhaps due to a typo. Is the first line supposed to specify the DM loss and not the GFM loss? Currently they are both statements about GFM.**
>
> The GFM loss with independent coupling is equal to the DM loss up to a time-dependent scalar multiplicative factor, so they have identical relative loss scales. We have clarified this point in the revised paper (**lines 249-251, 1104-1111**) and used DM loss as the reviewer suggests.

---

> ### Author Response · Authors · 2024-11-20
> **Reply to Reviewer 3r18 (Part 3/4)**
>
> > **[W4] These analyses are mostly limited to the image domain, and it's hard to understand how any of these (apply to other domains?)**
>
> While our suggested improvements have been empirically validated only on the image-domain, our suggestions are not image-modality specific and so similarly to prior works, we hope these improvements generalize to other domains, except for the a high-pass filter (HPF) in the loss function, which is image or audio specific.
>
> For example, seminal work of Karras 2022 [1] was demonstrated to work on image datasets (we use the same image datasets) and has since been extended to domains such as proteins see e.g. alphafold 3 [2].
>
> Indeed, in our reply to [W1], none of our discussion is confined to the image domain; or perhaps audio domain. Even for HPF, our theory allows the user to choose any invertible linear map $\phi$ in the loss function without losing the theoretical guarantees of flow matching. Hence, a user could design a suitable $\phi$ for learning ReFlow models on other domains. For instance, as the Reviewer mentioned in the Strengths section, if the user knows which frequency regimes are important, the user could combine a band pass filter for the particular frequency regime with the identity map to create $\phi$.
>
> [1] Karras et al., “Elucidating the Design Space of Diffusion-Based Generative Models”, NeurIPS, 2022.
>
> [2] Abramson, J., Adler, J., Dunger, J. et al. Accurate structure prediction of biomolecular interactions with AlphaFold 3. Nature 630, 493–500 (2024).
>
> > **[W5] Can the authors address how large the bias is in rectification under the different procedures somehow? By this, the reviewer means that each iterate of rectification introduces a bias unless the flow was initially learned perfectly. The reviewer understands this to be difficult to measure, but it would be nice to understand how these ablations affect or relate to the extent of this bias. This might sit better in the questions section.**
>
> The ReFlow procedure entails iteratively training parameterized velocity fields between marginal distributions. Training such a flow should preserve marginals, and sampling from the flow induces a new coupling between the same marginals, to be used for the next ReFlow iteration.
>
> One may train flows from scratch at each initialization, so any error accumulation will come from imperfect simulated marginals rather than bias in networks.  Our work aims to minimize error accumulation by improving training; we then further attempt to “correct” marginals both with projection and using a mixture of forward and backward pairs.
>
> The extent of the bias and its reduction with our techniques can be calculated by comparing Tables 6, 7, 8 in our main paper. Specifically, since we measure the performance of our ReFlow models via Frechet Inception Distance (Wasserstein-2 distance between Gaussian approximations of true and model distributions in the feature space of the Inception network), we may use (FID after ReFlow) - (FID of the diffusion model) as a proxy for the amount of bias introduced by ReFlow. Below, we summarize the extent of bias as we add improved training dynamics (DYN), improved learning (LRN), improved inference (INF), and guidance (GD) to the baseline setting. With everything combined, our techniques achieve a significant reduction in bias. We have added this discussion to **Appendix F.3** of our revised paper.
>
> | | **Baseline** | **DYN** | **DYN+LRN** | **DYN+LRN+INF** | **DYN+LRN+INF+GD** |
> |--|--|--|--|--|--|
> | **CIFAR10** |0.86|0.61|0.41|0.26|0.01|
> | **AFHQv2** |0.91|0.59|0.51|0.34|-0.05|
> | **FFHQ** |1.89|1.30|0.74|0.45|0.28|
>
> Table 1. Amount of bias introduced by ReFlow under different settings. Negative bias means our model achieves a better FID than
> the diffusion model used as $\mathbb{Q}_{01}^1$.
>
> Concurrent work (public after submission) performs an analysis of error accumulation [1, page 38] in a simplified setting and proves the forward/backward simulation of coupling (which we also use) results in bounded error accumulation.
>
> [1] Bortoli et al., Schrödinger Bridge Flow for Unpaired Data Translation, arXiv, 2024.
>
> > **[W6] The fundamental idea of FM and rectified flow are equivalent to the method of stochastic interpolants, and should probably be cited accordingly.**
>
> Thank you for pointing out the missing reference. We agree [1,2] were concurrent to other flow-matching / rectified flow papers. While we have cited [1] in line 901, we have further cited [1,2] in the introduction within our revised submission (**lines 38-39**).
>
> [1] Albergo et al., “Stochastic Interpolants: A Unifying Framework for Flows and Diffusions”
>
> [2] Albergo and Vanden-Eijnden, "Building Normalizing Flows with Stochastic Interpolants"

---

> ### Author Response · Authors · 2024-11-20
> **Reply to Reviewer 3r18 (Part 4/4)**
>
> > **[W7] There is in a sense a competing small set of remarks about the benefits of rectification vs distillation in the appendix section C, saying that there are benefits of rectification over distillation because rectification maintains an ODE-like structure. But this ignores the fact that the learning problem after a rectification is a distillation problem...and it too is now fitting to an imperfect teacher model. I'm not sure this remark really contributes much to the discussion.**
>
> We agree with the reviewer and argue that ReFlow can be viewed as a form of distillation in a broad sense and note this in **lines 807-808** of our original submission:  _“distillation and ReFlow are similar in the aspect that they train a new model using a teacher diffusion model”_
>
> We re-word this to be complementary instead of orthogonal: “we emphasize that they are, in fact, complementary approaches, and can benefit from one another” in the revised version. We also make the distinction we are discussing ReFlow vs discrete time distillation; given the broad interpretation of distillation.
>
> Whilst distillation is not the primary aim of our work, we include a discussion in the appendix (and not the main paper) to highlight both the similarities and differences to prior discrete time distillation approaches. We feel ReFlow is quite different from common diffusion distillation approaches and believe it is valuable to the community to include this discussion.
>
> Unlike typical discrete time diffusion distillation approaches such as Progressive Distillation, or adversarial training, which train on a discretisation of simulated data, ReFlow entails continuous time training whereby one of the marginal samples is simulated; hence outputs a valid continuous time neural ODE. The fact that the learnt model defines a continuous time ODE is important as it enables use of ODE solvers; likelihood evaluation and amenable to any other further distillation. Indeed [1] shows on CIFAR10 that one-step generative models distilled from ReFlow models outperform students distilled from diffusion models. [2] shows that the same holds on higher resolution (e.g, 512x512 or 1024x1024) datasets.
>
> The coupling used for training can be 1) sampled Gaussian and simulated data (e.g. images) from the previous ReFlow iteration ODE or 2) real images and simulated noise from the inverted ODE (again from the previous ReFlow step); or indeed a mixture of those as used in our paper.
>
> Note: [3] (public on October 14th 2024, after submission) introduces continuous time distillation via consistency models; also highlighting the disadvantages of discrete time distillation.
>
> [1] Liu et al., “Flow Straight and Fast: Learning to Generate and Transfer Data with Rectified Flow”, 2022.
>
> [2] Liu et al., “InstaFlow: One Step is Enough for High-Quality Diffusion-Based Text-to-Image Generation”, ICLR, 2024.
>
> [3] Lu and Song, "Simplifying, Stabilizing and Scaling Continuous-Time Consistency Models", 2024.
>
> > **[Q1] The reviewer is trying to understand the language of the following claim in 3.3.1: "Our improvement. We observe that learning the OT ODE is a harder task than learning the diffusion probability flow ODE. For instance, at t = 1, the optimal DM denoiser only needs to predict the data mean E[x0] for any input x1 ~ P1, but an optimal ReFlow denoiser has to directly map noise to image along the bijective OT map." Should the reviewer interpret this as being the claim that: The velocity field solving the probability flow is generically for E[dot(xt) | xt = x], and for a deterministic coupling P(x0,x1) = P0(x0) delta(x - T(x0)), this conditional expectation collapses because only one xt passes through x? This would be true for any monge coupling, and not just the OT coupling.**
>
> Yes, we imagine there would be (relatively) more difficulty for to train a flow for any deterministic ODE coupling; and indeed we agree with the reviewer predicting $\mathbb{E}[x_0 | x_t = x]$ becomes equivalent to predicting $x_0$ due to the deterministic coupling.

---

> ### Author Response · Authors · 2024-11-25
> **Have we addressed your concerns and questions?**
>
> As the discussion deadline is fast approaching, please let us know if there are any outstanding concerns we can address, or if you have any reservations preventing you from increasing the score. In particular, we have resolved any ambiguities regarding OT ODE vs. straight ODE in our revised paper, so your timely feedback would be highly appreciated.

---

### Official Review · Reviewer_7gZz · 2024-11-04

**Soundness:** 3
**Presentation:** 3
**Contribution:** 3
**Rating:** 8
**Confidence:** 4

**Summary:**

The paper studies ways of improving the ReFlow procedure for diffusion and flow matching. They break the design space into training dynamics, learning and inference, and show isolated ablations w.r.t. the choices here (weighting, time distribution, loss function, initialization, dropout, sampling, ODE solvers, [0, 1] discretization). The main experimental results ablate these on CIFAR10, AFHQv2, and FFHQ.

**Strengths:**

I find the paper well-presented. The design space is clearly communicated and the ablations are convincing. These dimensions show how to squeeze the performance on these standard image generation settings.

**Weaknesses:**

I do not find many weaknesses in the paper, as it clearly communicates the goal and setting and explores it nicely:

1. The experimental settings of CIFAR10, AFHQv2, and FFHQ are interesting, but only a small subset of all of the other potential uses and larger applications of flow matching and diffusion models. It would be interesting to see if the improvements continue holding in these.
2. The paper is niche in the sense that it relies on wanting to do generative modeling with ReFlow, and discussing the relevant dimensions here.

**Questions:**

None

---

> ### Author Response · Authors · 2024-11-20
> **Reply to Reviewer 7gZz**
>
> We would like to thank Reviewer 79Zz for the encouraging comments and the constructive review. Below, we address the Reviewer's concerns.
>
> > **[W1] The experimental settings of CIFAR10, AFHQv2, and FFHQ are interesting, but only a small subset of all of the other potential uses and larger applications of flow matching and diffusion models. It would be interesting to see if the improvements continue holding in these.**
>
> We agree with the reviewer that it would be interesting to explore the use of our proposed improvements in larger scale experiments and in other domains. Our largest scale was ImageNet-64, in pixel-space, which if performed in latent space using standard x8 compression rates would correspond to resolution 512.
>
> We are looking into extending our work in a number of directions such as larger scale image experiments similar to [1], and hope others will also use our suggested improvements in domain specific applications. Similar to how the work of Karras 2022 [3] uses the same datasets (cifar10, afhqv2, ffhq, imagenet-64) and has since been built for many large scale and diverse experiments, notably for higher resolution image modeling [4] or large scale protein design (alphafold3) [5].
>
> Larger applications of flow matching and diffusion models build on top of unconditional (in the sense that the prior is Gaussian) generative models. For instance, DDIB [6] for unpaired image-to-image translation concatenates two unconditional diffusion models to translate data from one domain to another. DPS [7] for inverse imaging also uses an unconditional diffusion model prior to guide an image restoration process. Hence, we are confident that our findings can directly translate to performance or efficiency improvements on several such downstream tasks in future work.
>
> [1] Liu et al., “InstaFlow: One Step is Enough for High-Quality Diffusion-Based Text-to-Image Generation”, ICLR, 2024.
>
> [2] Esser et al., “Scaling Rectified Flow Transformers for High-Resolution Image Synthesis”, ICML, 2024.
>
> [3] Karras et al., “Elucidating the Design Space of Diffusion-Based Generative Models”, NeurIPS, 2022.
>
> [4] Karras et al., “Analyzing and Improving the Training Dynamics of Diffusion Models”, CVPR, 2024.
>
> [5] Abramson, J., Adler, J., Dunger, J. et al. Accurate structure prediction of biomolecular interactions with AlphaFold 3. Nature 630, 493–500 (2024).
>
> [6] Su et al., “Dual Diffusion Implicit Bridges for Image-to-Image Translation”, ICLR, 2023.
>
> [7] Chung et al., “Diffusion Posterior Sampling for General Noisy Inverse Problems”, ICLR, 2023.
>
> > **[W2] The paper is niche in the sense that it relies on wanting to do generative modeling with ReFlow, and discussing the relevant dimensions here.**
>
> While the use of ReFlow is not (yet) commonplace, the problem that we tackle -- accelerating generative neural ODEs -- is fundamental. We show that with careful design choices, we can achieve significantly better performance metrics than prior neural ODEs (see Table 8), which suggests ReFlow is a strong contender amongst methods to accelerate generative models.

---

> > ### Comment · Reviewer_7gZz · 2024-11-20
> >
> > Thank you for the response. I've read through the other parts of the discussion and will maintain my original accept score for now

---

> > > ### Author Response · Authors · 2024-11-26
> > > **Many thanks for reading our rebuttal.**
> > >
> > > We would like to take this opportunity to thank you for reading our rebuttal and for raising very valid points. We remain available for further questions until the end of the rebuttal period.

---

### Author Response · Authors · 2024-11-25
**General Reply to All Reviewers**

We again thank the Reviewers for their time and the constructive feedback. We are glad that the Reviewers **“did not find many weaknesses in the paper”** (7gZz), **“acknowledges the size of the experimental undertaking”** (3r18), and found our paper to be **“well-written with clear ablations”** (Zbb5), and do **“a very good job at ablating design choices”** with **“state-of-the-art results”** so it may be **“very helpful for practitioners that face similar choices when training their model”** (VmAn). As the discussion deadline is approaching, please let us know if there are any remaining concerns.

---

### Meta-Review · Area_Chair_uSu1 · 2024-12-22

**Metareview:**

This paper focuses on improving the performance of ReFlow models for image generation while preserving their theoretical underpinnings. ReFlow aligns ODE trajectories by flow-matching marginal distributions. Despite its theoretical potential to perfectly translate between marginals, practical implementations often exhibit degraded sample quality, with prior approaches relying on heuristic fixes that compromise theoretical consistency.

The authors categorize their contributions into three dimensions: training dynamics, learning strategies, and inference techniques. They propose principled refinements, including loss reweighting, improved initialization from pre-trained models, optimized pair selection, and better ODE discretization strategies. Through extensive ablation studies on CIFAR10, FFHQ, and AFHQv2, they demonstrate that these improvements yield some gains in sample quality and FID scores.

While the paper has notable strengths, it also has several weaknesses: (1) Although many practical techniques are proposed, the paper lacks a deeper analysis of why these methods improve performance; (2) several reviewers highlighted inaccuracies in the paper’s understanding and claims related to OT; and (3) some proposed techniques result in only marginal performance improvements.

Despite these limitations, the paper’s contributions outweigh its shortcomings. All reviewers recommended acceptance, with several assigning high scores, reflecting confidence in its significance. Therefore, I recommend accepting this paper.

**Additional Comments On Reviewer Discussion:**

During the rebuttal period, most reviewers initially expressed positive opinions about the paper, with several noting that they did not identify significant weaknesses. The majority of the questions and comments raised were aimed at clarifying certain aspects of the paper or improving its presentation rather than pointing out substantial issues. The authors provided satisfactory responses that addressed these points effectively, reinforcing the reviewers’ consensus.

---

### Decision · Program_Chairs · 2025-01-22

Accept (Poster)